# Reinforcement Learning for Node Selection in Branch-and-Bound

**Alexander Mattick**          *alexander.mattick@iis.fraunhofer.de*
**Christopher Mutschler**      *christopher.mutschler@iis.fraunhofer.de*
*Fraunhofer IIS, Fraunhofer Institute for Integrated Circuits IIS*
*Nordostpark 84, 90411 Nürnberg, Germany*

**Reviewed on OpenReview:** *https: // openreview. net/ forum? id= VrWl6yNk1E*

## Abstract

A big challenge in branch-and-bound lies in identifying the optimal node within the search tree from which to proceed. Current state-of-the-art selectors utilize either hand-crafted ensembles that automatically switch between naive subnode selectors, or learned node selectors that rely on individual node data. In contrast to existing approaches that only consider isolated nodes, we propose a novel simulation technique that uses reinforcement learning (RL) while considering the entire tree state. To achieve this, we train a graph neural network that produces a probability distribution based on the path from the model's root to its "to-be-selected" leaves. Representing node-selection as a probability distribution allows us to train a decision-making policy using state-of-the-art RL techniques that capture both intrinsic node-quality and node-evaluation costs. Our method induces a high quality node selection policy on a set of varied and complex problem sets, despite only being trained on specially designed synthetic traveling salesmen problem (TSP) instances. Using such a *fixed pretrained* policy shows significant improvements on several benchmarks in optimality gap reductions and per-node efficiency under a short time limit of 45s and demonstrates generalization to a significantly longer 5min time limit.

## 1 Introduction

The optimization paradigm of mixed integer programming plays a crucial role in addressing a wide range of complex problems, including scheduling (Bayliss et al., 2017), process planning (Floudas & Lin, 2005), and network design (Menon et al., 2013). A prominent algorithmic approach employed to solve these problems is *branch-and-bound (BnB)*, which recursively subdivides the original problem into smaller sub-problems through variable branching and pruning based on inferred problem bounds. BnB is also one of the main algorithms implemented in SCIP (Bestuzheva et al., 2021), a state-of-the art mixed integer linear and mixed integer nonlinear solver.

An often understudied aspect is the node selection problem, which involves determining which nodes within the search tree are most promising for further exploration. This is due to the intrinsic complexity of understanding the emergent effects of node selection on overall performance for human experts. Contemporary methods addressing the problem of node selection typically adopt a perspective per node (Yilmaz & Yorke-Smith, 2021; He et al., 2014; Morrison et al., 2016), incorporating varying levels of complexity and relying on imitation learning (IL) from existing heuristics (Yilmaz & Yorke-Smith, 2021; He et al., 2014). However, they fail to fully capture the rich structural information present within the branch-and-bound tree itself.

We propose a novel selection heuristic that leverages the power of bi-simulation: The BnB tree structure and its expansion/pruning dynamics are directly replicated inside our neural network model. We now employ reinforcement learning (RL) to learn a heuristic that is naturally well aligned to the branch-and-bound problem, see Fig. 1. The resulting method is able to take advantage of the inherent structure within the

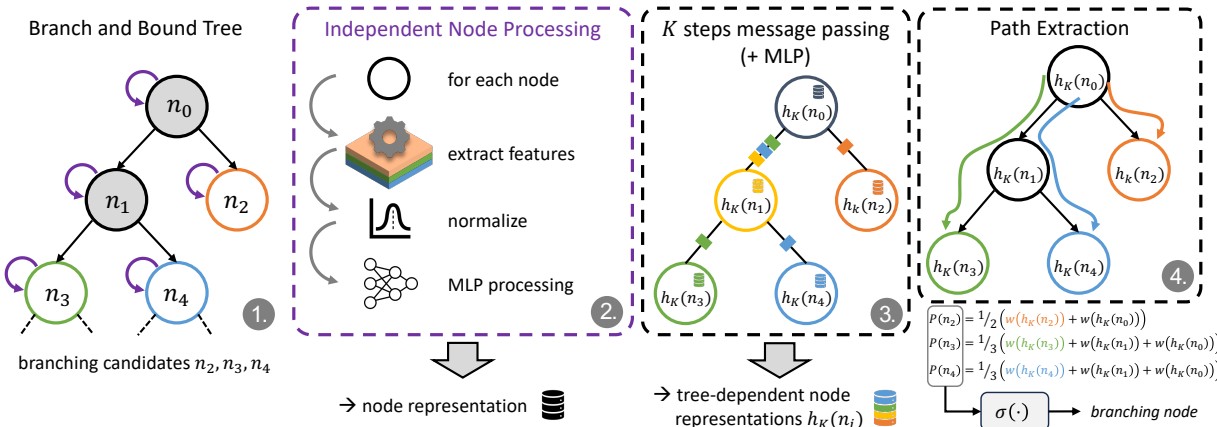

Figure 1: Overview of our approach: (1) SCIP solves individual nodes and executes existing heuristics. (2) Features are extracted from every branch-and-bound node and sent to individual normalization and embedding. (3) The node embeddings are subject to $K$ steps of GNN message passing on the induced tree-structure. The embeddings are projected to scalar weights and values using a Multi Layer Perceptron (MLP). (4) Based on the node embeddings, we generate root-to-leave paths, from which we sample the next node. The resulting node is submitted to SCIP and we return to step 1.

branch-and-bound algorithm, leading to a superior generalization compared to the seminal work by Labassi et al. (2022). This allows the RL policy to directly account for the BnB tree's dynamics.

We reason that RL specifically is a good fit for this type of training as external parameters outside the pure quality of a node have to be taken into account. For example, a node $A$ might promise a significantly bigger decrease in the expected optimality gap than a second node $B$, but node $A$ might take twice as long to evaluate, making $B$ the "correct" choice despite its lower theoretical utility. By incorporating the bi-simulation technique, we can effectively capture and propagate relevant information throughout the tree.

The main contributions of this paper are:

1. a method for processing branch-and-bound trees of arbitrary size, which probabilistically samples leaves of the BnB tree by modeling the selection as a root-to-leaf traversal of the BnB tree,
2. a new way of sampling optimization instances of intermediary difficulty,
3. a thorough time-limited benchmark on industrial-scale instances contained in TSPLIB, MINLPLib, and MIPLIB, and
4. an analysis of the learned node selector using explainable-AI techniques

To the best of our knowledge, we are the first to present a learned selection method that now only generalizes across instance types but that is also applicable to nonlinear mixed-integer optimization.

The rest of this article is structured as follows. Sect. 2 describes branch-and-bound and Sect. 3 discusses related work. Sect. 4 explains our method, i.e., our node representation, our node selection agent, and how we generate training instances. Sect. 5 discusses the experiments, and Sect. 6 discusses limitations. Sec. 7 concludes.

## 2   Branch and Bound

BnB is one of the most effective methods for solving mixed integer programming (MIP) problems. It recursively solves relaxed versions of the original problem, gradually strengthening the constraints until it finds an optimal solution. The first step relaxes the original MIP instance into a tractable subproblem by dropping all integrality constraints such that the subproblem can later be strictified into a MIP solution. For simplicity, we focus our explanation to the case of mixed integer linear programs (MILP) while our method theoretically works for any type of constraint allowed in SCIP (see nonlinear results in Sec. 5.3.1,

and Bestuzheva et al. (2023)). Concretely a MILP has the form

$$P_{\text{MILP}} = \min\{c_1^T x + c_2^T y | Ax + By \geq b, y \in \mathbb{Z}^n\}, \tag{1}$$

where $c_1$ and $c_2$ are coefficient vectors, $A$ and $B$ are constraint matrices, and $x$ and $y$ are variable vectors. The integrality constraint $y \in \mathbb{Z}^n$ requires $y$ to be an integer. In the relaxation step, this constraint is dropped, leading to the following simplified problem:

$$P_{\text{relaxed}} = \min\{c_1^T x + c_2^T y | Ax + By \geq b\}. \tag{2}$$

Now, the problem becomes a linear program without integrality constraints, which can be exactly solved using the Simplex (Dantzig, 1982) or other efficient linear programming algorithms.

After solving the relaxed problem, BnB proceeds to the branching step: First, a non-integral $y_i$ is chosen. The branching step then derives two problems: The first problem (Eq. 3) adds a lower bound to variable $y_i$, while the second problem (Eq. 4) adds an upper bound to variable $y_i$. These two directions represent the rounding choices to enforce integrality for $y_i$:[1]

$$P_{\text{relaxed}}^1 = \min\{c_1^T x + c_2^T y | Ax + By \geq b, y_i \leq \lfloor c \rfloor\} \tag{3}$$
$$P_{\text{relaxed}}^2 = \min\{c_1^T x + c_2^T y | Ax + By \geq b, y_i \geq \lceil c \rceil\} \tag{4}$$

The resulting decision tree, with two nodes representing the derived problems can now be processed recursively. However, a naive recursive approach exhaustively enumerates all integral vertices, leading to an impractical computational effort. Hence, in the bounding step, nodes that are deemed to be worse than the currently known best solution are discarded. To do this, BnB stores previously found solutions which can be used as a lower bound to possible solutions. If a node has a lower bound larger than a currently found integral solution, no node in that subtree has to be processed.

The interplay of these three steps—relaxation, branching, and bounding—forms the core of branch-and-bound. It enables the systematic exploration of the solution space while efficiently pruning unpromising regions. Through this iterative process, the algorithm converges towards the globally optimal solution for the original MIP problem, while producing optimality bounds at every iteration.

## 3    Related Work

Learning node selection, where learned heuristics pick the best node to continue from the search tree, has only rarely been addressed in research. Prior work that learns such node selection strategies made significant contributions to improve the efficiency and effectiveness of the optimization.

Notably, many approaches rely on per-node features and Imitation Learning (IL). Otten & Dechter (2012) study the estimation of subproblem complexity as a means to enhance parallelization efficiency. By estimating the complexity of subproblems, the algorithm can allocate computational resources of parallel solvers more effectively. Yilmaz & Yorke-Smith (2021) employ IL to directly select the most promising node for exploration. Their approach utilizes a set of per-node features to train a model that can accurately determine which node to choose. He et al. (2014) make use of support vector machines and IL to create a hybrid heuristic based on existing heuristics. By leveraging per-node features, their approach aims to improve node selection decisions. While these prior approaches yield valuable insights, they are inherently limited by their focus on per-node features.

Labassi et al. (2022) propose the use of Siamese graph neural networks, representing each node as a graph that connects variables with the relevant constraints. Their objective is to directly imitate an optimal diving oracle, which descends depth-first towards a node containing a minimal value. This approach facilitates learning from node comparisons and enables the model to make informed decisions during node selection. This means that Labassi et al. (2022) manages to study pairs of leaf-subproblems, rather than studying every

---

[1]There are non-binary, "wide" branching strategies which we will not consider here explicitly. However, our approach is flexible enough to allow for arbitrary branching width. See Morrison et al. (2016) for an overview.

subproblem in isolation. However, this method still does not utilize the high information content given by the tree structure itself: The search tree itself slowly gathers information in the form of, e.g., cutting planes or the optimality of solutions close to (but not inside) the current subproblem's feasible set.

Aside of the work in learning node selection, there is also substantial work in learning different heuristics for branch and bound. One commonly learned component of branch-and-bound is the *variable selection* heuristic (Scavuzzo et al., 2022; Parsonson et al., 2022; Gasse et al., 2019; Etheve et al., 2020). Variable selection is the problem of which variable to apply the branch for the subproblems, i.e., which $y_i$ to add a constraint to in Eqs 3 and 4. Other frequently learned heuristics include *cut-selection* methods (Tang et al., 2019; Wang et al., 2023; Paulus et al., 2022; Turner et al., 2022). Cut selection is part of an extension of branch-and-bound known as branch-and-cut, which additionally adds constraints to sub-problems that remove solutions that are feasible in a node's linear relaxation, but are infeasible in the original MILP model. Specifically, if $x$ is a solution to the relaxed Equation 2, but not a solution to the MILP Equation 1, a new constraint is placed into the relaxation that removes the found $x$ from the problem. Such cuts are generally not unique. Cut selection is the process of picking the best cuts from a set of different feasible cuts. *However, all these (types of) heuristics are orthogonal to our node selection problem: One can apply both our learned node selector as well as any combination of cut and variable selectors.*

Compared to both cut and variable selection, node selection is severely under-researched due to its inherent complexity. While both variable selection and cut selection can be solved by studying a single sub-problem in isolation, node selection has to study the set of all possible sub-problems as a whole, which increases the complexity of building node selectors.

## 4  Methodology

We model the node selection process as a sequential Markov Decision Process (MDP). An MDP consists of a set of observable states $s \in \mathcal{S}$, a set of controllable actions $a \in \mathcal{A}$, a reward function $R(s)$ describing the quality of an individual state $s \in \mathcal{S}$ in isolation, and a (typically unknown) transition distribution $T : \mathcal{S} \times \mathcal{A} \to \mathcal{S}$ that takes in a current state $s \in \mathcal{S}$ and an action $a \in \mathcal{A}$, proposed by a decision-making policy $\pi : \mathcal{S} \to \mathcal{A}$, and produces a distribution over new state $s' \in \mathcal{S}$. The policy is usually thought of as a conditional distribution $\pi(a|s)$ producing the next action $a \in \mathcal{A}$ under the knowledge of the current state $s \in \mathcal{S}$. Our objective is to find the policy optimal $\pi^*$, which maximizes the expected discounted sum of future rewards:

$$\pi^* = \text{argmax}_\pi \, \mathbb{E}\left[\sum_{t=0}^{t_{\max}} \gamma^t R(s_t)\pi(a_t|s_t)T(s_{t+1}|s_t, a_t)\right], \tag{5}$$

where $0 < \gamma < 1$ is a factor to trade immediate over future rewards (Sutton & Barto, 2018).

RL is the process of solving such MDPs by repeatedly rolling out $\pi$ on an environment and using the resulting trajectories to find a policy $\pi$ that maximizes the return. Several solvers for MDPs exist, such as Deep Q-Networks (DQN) (Mnih et al., 2015) or Proximal Policy Optimization (PPO) (Schulman et al., 2017). In this paper we use PPO to solve the MDP.

In our case, we phrase our optimization problem as an MDP, where states $s_t$ are represented using directed branch-and-bound trees, actions $a_t$ are all selectable leaf-nodes, and the reward $R$ is set according to Eq. 6 in Sec. 4.1. The transitions are given by the SCIP-solver and its heuristics: Given the current state (BnB tree) and the node selection by $\pi$, the SCIP solver produces new nodes as children of the selected ones, and prunes unnecessary nodes from the tree. To solve the MDP, PPO needs a representation for the policy $\pi$ and a model describing the state-value-function, defined as $V(s) = \max_a Q(s_t, a)$ (see Schulman et al. (2017)).

In our model, we represent the branch-and-bound tree as a directed Graph Neural Network (GNN) whose nodes and edges are in an 1:1 correspondence to the branch-and-bound nodes and edges, see Sec. 4.2. This means that for every node in the BnB tree there is exactly one node in the GNN, and every connection in the BnB tree is replicated in the GNN graph. As the BnB tree grows, the GNN graph grows as well, solving one of the biggest issues in node selection: Since the BnB tree grows and shrinks dynamically, the number of different node selections (= actions) changes drastically. One benefit of our GNN model is that the resulting

directed graph automatically grows/shrinks the action space as new nodes are added/pruned from the tree since the policy's action-space directly corresponds to the leaves of the GNN/BnB tree. This allows our network to phrase node selection explicitly as a distribution over node-selections, rather than using implicit comparisons as in Labassi et al. (2022).

### 4.1 Reward Definition

A common issue in MIP training is the lack of good reward functions. For instance, the primal-dual integral used in Gasse et al. (2022) has the issue of depending on the scale of the objective, needing a primally feasible solution at the root, and being very brittle if different instance difficulties exist in the instance pool. Gasse et al. (2022) circumvents these issues by generating similar instances, while also supplying a feasible solution at the root. However, to increase generality, we do not want to make as strict assumptions as having similar instance difficulties, introducing a need for a new reward function. A reasonable reward measure captures the achieved performance against a well-known baseline. In our case, we choose our reward, to be

$$r = - \left( \frac{gap(\textit{Node Selector})}{gap(\textit{SCIP})} - 1 \right), \tag{6}$$

where $gap(\textit{SCIP})$ denotes the optimality gap reachable by the standard SCIP node selector in the set time limit (for us 45s), and $gap(\textit{node selector})$ denotes the gap reachable by our method within the set time limit. This represents the relative performance of our selector compared to the performance achieved by SCIP.

Intuitively, the aim of this reward function is to decrease the optimality gap achieved by our node-selector (i.e., $gap(\textit{node selector})$), normalized by the results achieved by the existing state-of-the-art node-selection methods in the SCIP (Bestuzheva et al., 2021) solver (i.e., $gap(\textit{SCIP})$). This accounts for varying instance hardness in the dataset. Further, we shift this performance measure, such that any value $> 0$ corresponds to our selector being superior to existing solvers, while a value $< 0$ corresponds to our selector being worse, and clip the reward between $(1, -1)$ to ensure symmetry around zero as is commonly done in prior work (see, e.g. Mnih et al. (2015); Dayal et al. (2022)). This formulation has the advantage of looking at relative improvements over a baseline, rather than absolute performance, which allows us to use instances of varying complexity during training.

We also experimented with other metrics that aim to normalize for the complexity of the instance themselves (see Appendix C). We found that the reward metric as proposed here is the most stable to train and yields the best results w.r.t. minimizing the optimality gap.

### 4.2 Tree Representation

To represent the MDP's state, we propose a novel approach that involves bi-simulating the existing branch-and-bound tree using a graph neural network (GNN). Specifically, we consider the state to be a directed tree $T = (V, E)$, where $V$ are individual subproblems created by SCIP, and $E$ are edges set according to SCIP's node-expansion rules. More precisely, the tree's root is the principle relaxation (Eq. 2), and its children are the two rounding choices (Eqs. 3 and 4). BnB proceeds to recursively split nodes, adding the splits as children to the node it was split from.

For processing inside the neural network, we extract features (see Appendix A) from each node, keeping the structure intact: $T = (\text{extract}(V), E)$. We ensure that the features stay at a constant size, independent from, e.g., the number of variables and constraints, to enable efficient batch-evaluation of the entire tree.

For processing the tree $T$, we use message-passing from the children to the parent. Pruned or missing nodes are replaced with a constant to maintain uniformity over the graph structure. Message-passing enables us to consider the depth-$K$ subtree under every node by running $K$ steps of message passing from the children to the parent. Concretely, the internal representation can be thought of initializing $h_0(n) = x(n)$ (with $x(n)$ being the features associated with node $n$) and then running $K$ iterations jointly for all nodes $n$:

$$h_{t+1}(n) = h_t(n) + \text{emb} \left( \frac{h_t(\text{left}(n)) + h_t(\text{right}(n))}{2} \right), \tag{7}$$

where $\mathrm{left}(n)$ and $\mathrm{right}(n)$ are the left and right children of $n$, respectively, $h_t(n)$ is the hidden representation of node $n$ after $t$ steps of message passing, and emb is a function that takes the mean hidden state of all embeddings and creates an updated node embedding. Doing this for $K$ steps aggregates the information of the depth-K limited subtree of $n$ into the node $n$.

## 4.3 RL for Node Selection

While the GNN model is appealing, it is impossible to train using contemporary imitation learning techniques, as the expert's action domain (i.e., leaves) may not be the same as the policy's action domain, meaning that the divergence between these policies is undefined.

To solve this problem, we choose to use model-free reinforcement learning techniques to directly maximize the probability of choosing the correct node. State-of-the-art reinforcement learning techniques, such as proximal policy optimization (PPO) (Schulman et al., 2017), need to be able to compute two functions: The value function $V(s) = \max_{a \in \mathcal{A}} Q(s, a)$, and the likelihood function $\pi(a|s)$. PPO uses the value function to reduce the variance of the advantage computation. In our work, we utilize the Generalized Advantage Estimator (GAE) Schulman et al. (2015) which provides a continuous mixture of bootstrapped and Monte Carlo estimated advantages. As it turns out, there are efficient ways to compute both of these values from a tree representation:

Firstly, we can produce a probability distribution of node-selections (i. e., our actions) by computing the expected "weight" across the unique path from the graph's root to the "to-be-selected" leaves. These weights are mathematically equivalent to the unnormalized log-probabilities of choosing a leaf-node by recursively choosing either the right or left child with probability $p$ and $1 - p$ respectively. The full derivation of this can be found in Appendix B. Concretely, let $n$ be a leaf node in the set of choosable nodes $\mathcal{C}$, also let $P(r, n)$ be the unique path from the root $r$ to the candidate leaf node, with $|P(r, n)|$ describing its length. We define the expected path weight $W'(n)$ to a leaf node $n \in \mathcal{C}$ as

$$W'(n) = \frac{1}{|P(r, n)|} \sum_{u \in P(r,n)} W(h_K(u)). \tag{8}$$

Selection now is performed in accordance to sampling from a selection policy $\pi$ induced by

$$\pi(n | \text{tree}) = \mathrm{softmax}\left(\{W'(n) | \forall n \in \mathcal{C}\}\right). \tag{9}$$

Intuitively, this means that we select a node exactly if the expected utility along its path is high. Note that this definition is naturally self-correcting as erroneous over-selection of one subtree will lead to that tree being completed, which removes the leaves from the selection pool $\mathcal{C}$.

Similar to the log-likelihood, we parameterize the value function as a root-to-leaf sum, which is aggregated using a $\max(\cdot)$ over paths.

$$f(n|s) = \frac{\tilde{f}(n|s)}{|P(r, n)|} \tag{10}$$

$$\tilde{f}(n|s) = \tilde{f}(\text{left}|s) + \tilde{f}(\text{right}|s) + f(h_K(n)|s) \tag{11}$$

$$V(s) = \max\left\{f(n) \mid \forall n \in \mathcal{C}\right\}, \tag{12}$$

where $f(h_n)$ is a trained per-node estimator, $\hat{f}$ is the path-aggregated $f$-function, and $\mathcal{C}$ is the set of open nodes as proposed by the branch-and-bound method. For an alternative parametrization of $V(\cdot)$ that justifies the use of $\tilde{f}$ as a Q-function, see Appendix J.

Since we can compute the action likelihood $\pi(a|s)$, the value function $V(s)$, and the Q-function $Q(s, a)$, we can use any Actor-Critic method (like PPO (Schulman et al., 2017) or A3C (Mnih et al., 2016)) for training this model. We use PPO due to its ease of use and robustness.

According to these definitions, we only need to predict the node embeddings for each node $h_K(n)$, the per-node q-function $q(h_K(n)|s)$, and the weight of each node $W(h_K(u))$. We parameterize all of these as MLPs (more architectural details can be found in Appendix E).

This method provides low, but measurable overhead compared to existing node selectors, even if we discount the fact that our Python-based implementation is vastly slower than SCIP's highly optimized C-based implementation. Hence, we focus our model on being efficient at the beginning of the selection process, where good node selections are exponentially more important as finding more optimal solutions earlier allows to prune more nodes from the exponentially expanding search tree. Specifically we evaluate our heuristic at every node for the first 250 selections, then at every tenth node for the next 750 nodes, and finally switch to classical selectors for the remaining nodes.[2]

### 4.4 Data Generation & Agent Training

In training MIPs, a critical challenge lies in generating sufficiently complex training problems. First, to learn from interesting structures, we need to decide on some specific problem, whose e.g. satisfiability is knowable as generating random constraint matrices will likely generate empty polyhedrons, or polyhedrons with many eliminable constraints (e.g., in the constraint set consisting of $c^T x \leq b$ and $c^T x \leq b + \rho$ with $\rho \neq 0$ one constraint is always eliminable). This may seem unlikely, but notice how we can construct aligned $c$ vectors by linearly combining different rows (just like in LP-dual formulations). In practice, selecting a sufficiently large class of problems may be enough as during the branch-and-cut process many sub-problems of different characteristics are generated. Since our algorithm naturally decomposes the problem into sub-trees, we can assume any policy that performs well on the entire tree also performs well on sub-polyhedra generated during the branch-and-cut.

For this reason we consider the large class of Traveling Salesman Problem (TSP), which itself has rich use-cases in planning and logistics, but also in optimal control, the manufacturing of microchips and DNA sequencing (see Cook et al. (2011)). The TSP problem consists of finding a round-trip path in a weighted graph, such that every vertex is visited exactly once, and the total path-length is minimal (for more details and a mathematical formulation, see Appendix F).

For training, we would like to use random instances of TSP but generating them can be challenging. Random sampling of distance matrices often results in easy problem instances, which do not challenge the solver. Consequently, significant effort is being devoted into devising methods for generating random but hard instances, particularly for problems like the TSP, where specific generators for challenging problems have been designed (see Vercesi et al. (2023) and Rardin et al. (1993)). However, for our specific use cases, these provably hard problems may not be very informative as they rarely contain efficiently selectable nodes.

To generate these intermediary-difficult problems, we adopt a multi-step approach: We begin by generating random instances and then apply some mutation techniques (Bossek et al., 2019) to introduce variations, and ensure diversity within the problem set. Next, we select the instance of median-optimality gap from the pool, which produces an instance of typical difficulty. The optimality gap, representing the best normalized difference between the lower and upper bound for a solution found during the solver's budget-restricted execution, serves as a crucial metric to assess difficulty. This method is used to produce 200 intermediary-difficult training instances.

To ensure the quality of candidate problems, we exclude problems with more than 100% or zero optimality gap, as these scenarios present challenges in reward assignment during RL. To reduce overall variance of our training, we limit the ground-truth variance in optimality gap. Additionally, we impose a constraint on the minimum number of nodes in the problems, discarding every instance with less than 100 nodes. This is essential as we do not expect such small problems to give clean reward signals to the reinforcement learner.

## 5 Experiments

For our experiments we consider the instances of TSPLIB (Reinelt, 1991) and MIPLIB (Gleixner et al., 2021) which are one of the most used datasets for benchmarking MIP frameworks and thusly form a strong baseline to test against. We further test against the UFLP instance generator by (Kochetov & Ivanenko, 2005), which

---

[2]This accounts for the "phase-transition" in MIP solvers where optimality needs to be proved by closing the remaining branches although the theoretically optimal point is already found (Morrison et al., 2016). Note that with a tuned implementation we could run our method for more nodes, where we expect further improvements.

specifically produces instances hard to solve for branch-and-bound, and against MINLPLIB (Bussieck et al., 2003), which contains *mixed integer nonlinear programs*, to show generalization to vastly different problems.

We perform all our training and the experiments on a Ryzen 7 5800x3d CPU, where training completes after approx. 6h. After the model is trained on the synthetic TSP instances, we freeze the network and apply it to both the "in distribution" TSPLIB and the "out of distribution" UFLP, MINLPLIB and MIPLIB instances. This is in contrast to prior work such as those from Labassi et al. (2022), who train separate models for all instance types and do not manage to generalize to foreign instances (see Sec 5.4). Our source code is publicly available and can be used to reproduce the experiments.[3] We use the default hyperparameters as proposed by CleanRL (Huang et al., 2021).

For our benchmarks on TSPLIB, UFLP, MIPLIB and MINLPLIB, we impose a time limit of 45s and 5min. We report the expected reward (Eq. 6), the geometric mean of the achieved gap for both our and the baseline method, and the win-rate of our method as defined by $\text{gap}(ours) \leq \text{gap}(baseline)$.[4] For longer runs with a timelimit of 30min see Appendix D.

## 5.1 Baselines

We run both our method and SCIP for 45s. We then filter out all runs where SCIP has managed to explore less than 5 nodes, as in these runs even perfect node selection makes no difference in performance. If we included those in our average, we would have a significant number of lines where our node-selector has zero advantage over the traditional SCIP one, not because our selector is better or worse than SCIP, but simply because it wasn't called in the first place. We set this time-limit relatively low as our prototype selector only runs at the beginning of the solver process, meaning that over time the effects of the traditional solver take over.

Additionally, we also consider an evaluation with a 5min time limit. For those runs, the limit until we switch to the naive node selector is set to 650 nodes to account for the higher time limit. In general, the relative performance of our method compared to the baseline increases with the 5min budget.

Finally, we also benchmark against the previous state-of-the-art by Labassi et al. (2022), which represents optimal node selection as a comparison problem where a siamese network is used to compare different leaf nodes against each other and picking the "highest" node as the next selection.

## 5.2 Results

While all results can be found in Appendix I we report an aggregated view for each benchmark in Table 1. In addition to our "reward" metric we report the winning ratio of our method over the baseline, and the geometric mean of the gaps at the end of solving (lower is better).

For benchmarking and training, we leave all settings, such as presolvers, primal heuristics, diving heuristics, constraint specializations, etc. at their default settings to allow the baseline to perform best. All instances are solved using the same model without any fine-tuning. We expect that tuning, e.g., the aggressiveness of primal heuristics, increases the performance of our method, as it decreases the relative cost of evaluating a neural network, but for the sake of comparison we use the default parameters for all our tests. We train our node selection policy on problem instances according to Sec. 4.4 and apply it on problems from different benchmarks.

---

[3]Source code: `https://github.com/MattAlexMiracle/BranchAndBoundBisim`

[4]All of these metrics give a slightly different view of the performance of our method: The **reward** metric is naturally balanced to weight the improvement relative to the difficulty of the problem as reward captures the improvement over the baseline in percent (higher is better). The downside of this is that because reward is balanced, it will assign the same reward to an improvement of 5gap → 2.5gap as to 0.5gap → 0.25gap, which can overemphasize the impact of easy to solve instances. **Win-rate** is a supplementary metric showing the percentage of instances where our model *beats the baseline*. This can help show how the improvements are distributed among the instances (higher is better). Finally, comparing the **geometric mean** is the gold-standard tool in measuring solver performance (Bestuzheva et al., 2021). In contrast to the reward metric, geometric means are specifically designed to reject outliers increasing the robustness of the metric. We report both our geometric mean and the SCIP baseline's geometric mean (our gap < SCIP gap is better).

Table 1: Performance across benchmarks (the policy only saw TSP instances during training). The reward is the normalized improvement over the baseline (Eq. 6) averaged over the benchmarking dataset, the win-rate shows how often our method produces a better (or in case of both methods reaching 0% gap equal) optimality gap, and the geometric means are the shifted geometric means of optimality gaps computed over the benchmarking dataset.

| Benchmark | Reward | Win-rate | geo-mean Ours | geo-mean SCIP |
|---|---|---|---|---|
| TSPLIB (Reinelt, 1991) | 0.184 ±0.52 | 0.50 | 0.931 | 0.957 |
| UFLP (Kochetov & Ivanenko, 2005) | 0.078 ±0.19 | 0.636 | 0.491 | 0.520 |
| MINLPLib (Bussieck et al., 2003) | 0.487 ±0.60 | 0.852 | 28.783 | 31.185 |
| MIPLIB (Gleixner et al., 2021) | 0.140 ±0.69 | 0.769 | 545.879 | 848.628 |
| TSPLIB@5min | 0.192 ±0.51 | 0.600 | 1.615 | 2.000 |
| MINLPlib@5min | 0.486 ±0.60 | 0.840 | 17.409 | 20.460 |
| MIPLIB@5min | 0.150 ±0.69 | 0.671 | 66.861 | 106.400 |

First, we will discuss TSPLIB itself, which while dramatically more complex than our selected training instances, still contains instances from the same problem family as the training set (Sec. 5.2.1). Second, we consider instances of the Uncapacitated Facility Location Problem (UFLP) as generated by Kochetov & Ivanenko (2005)'s problem generator. These problems are designed to be particularly challenging to branch-and-bound solvers due to their large optimality gap (Sec. I.1). While the first two benchmarks focused on specific problems (giving you a notion of how well the algorithm does on the problem itself) we next consider "'meta-benchmarks" that consist of many different problems, but relatively few instances of each. MINLPLIB (Bussieck et al., 2003) is a meta-benchmark for *nonlinear* mixed-integer programming (Sec. 5.3.1), and MIPLIB (Gleixner et al., 2021) a benchmark for mixed integer programming (Sec. 5.3.2). We also consider generalization against the uncapacitated facility location problem using a strong instance generator, see Appendix I.1. Our benchmarks are diverse and complex and allow to compare algorithmic improvements in state-of-the-art solvers.

### 5.2.1 TSPLIB

From an aggregative viewpoint we outperform the SCIP node selection by $\approx 20\%$ in reward. However, the overall "win-rate" is only 50% as the mean-reward is dominated by instances where our solver has significant performance improvements: When our method loses, it loses by relatively little (e.g., `att48`: ours 0.287 vs base 0.286), while when it wins, it usually does so by a larger margin.

Qualitatively, it is particularly interesting to study the problems our method still loses significantly against SCIP (in four cases). A possible reason why our method significantly underperforms on `Dantzig42` is that our implementation is just too slow, considering that the baseline manages to evaluate $\approx 40\%$ more nodes. A similar observation can be made on `eil51` where the baseline manages to complete 5× more nodes. `rd100` is also similar to `Dantzig` and `eil51` as the baseline is able to explore 60% more nodes. `KroE100` is the first instance our method loses against SCIP, despite exploring a roughly equal amount of nodes. We believe that this is because our method commits to the wrong subtree early and never manages to correct into the proper subtree. Ignoring these four failure cases, our method is either on par (up to stochasticity of the algorithm) or exceeds the baseline significantly.

It is also worthwhile to study the cases where both the baseline and our method hit 0 optimality gap. Both algorithms reaching zero optimality gap can be seen as somewhat of a special case, since soley looking at the solution value is insufficient to figure out which method performs better in practice. A quick glance at instances like `bayg29`, `fri26`, `swiss42` or `ulysses16` shows that our method tends to finish these problems with significantly fewer nodes explored. This is not captured by any of our metrics since those only look at solution quality, not the efficiency of reaching that solution. If the quality of the baseline and ours is the same, it makes sense to look at solution efficiency instead. Qualitatively, instances like `bayg29` manage to reach the optimum in only $\frac{1}{3}$ the number of explored nodes, which showcases a significant improvement in node-selection quality. It is worth noting that, due to the different optimization costs for different nodes, it

not always true that evaluating fewer nodes is faster in wall-clock time. In practice, "fewer nodes is better" seems to be a good rule-of-thumb to check algorithmic efficiency.

## 5.3 UFLP

The UFLP benchmark designed by Kochetov & Ivanenko (2005) is specifically built to be hard to solve by branch-and-bound solvers due to its large duality gap. Despite this, our method manages to outperform the baseline significantly. This is meaningful since this benchmark is a specially designed worst-case scenario: The fact that our method still outperforms the baseline provides good evidence of the efficacy of the method.

### 5.3.1 MINLPLIB

We now consider MINLPs. To solve these, SCIP and other solvers use branching techniques that cut nonlinear (often convex) problems from a relaxed master-relaxation towards true solutions. We consider MINLPLib (Bussieck et al., 2003), a meta-benchmark consisting of hundreds of diverse synthetic and real-world MINLP instances of varying different types and sizes. As some instances take hours to solve even a single node, we filter out all problems with fewer than 5 nodes, as the performance in those cases is independent of the node-selectors performance (Full results Appendix I.4).

Our method still manages to outperform SCIP, even on MINLPs, despite never having seen a single MINLP problem before, see Table 1. Qualitatively, our method either outperforms or is on par with the vast majority of problems, but also loses significantly in some problems, greatly decreasing the average. Studying the cases our method loses convincingly (see Appendix I.4), we find a similar case as in TSPLIB, where the baseline implementation is so much more optimized that significantly more nodes can get explored. We suspect the reason our method has the biggest relative improvements on MINLPs is due to the fact that existing e.g. pseudocost based selection methods do not perform as well on spatial-branch-and-bound tasks.

We expect features specifically tuned for nonlinear problems to increase performance by additional percentage points, but as feature selection is orthogonal to the actual algorithm design, we leave more thorough discussion of this to future work [5].

### 5.3.2 MIPLIB

Last, but not least we consider the meta-benchmark MIPLIB (Gleixner et al., 2021), which consists of hundreds of real-world mixed-integer programming problems of varying size, complexity, and hardness. Our method is either close to or exceeds the performance of SCIP, see Table 1.

Considering per-instance results, we see similar patterns as in previous failure cases: Often we underperform on instances that need to close many nodes, as our method's throughput lacks behind that of SCIP. We expect that a more efficient implementation alleviates the issues in those cases.

We also see challenges in problems that are far from the training distribution, specifically satisfaction problems. Consider `fhnw-binpack4-48`, were the baseline yields an optimality gap of 0 while we end at $+\infty$. This is due to the design of the problem: Instead of a classical optimization problem, this is a satisfaction problem, where not an optimal value, but *any* feasible value is searched, i.e., we either yield a gap of 0 (solution found), or a gap of $+\infty$ (solution not found), as no other gap is possible. Notably, these kinds of problems may pose a challenge for our algorithm, as the node-pruning dynamics of satisfying MIPs are different than the one for optimizing MIPs: Satisfying MIPs can only rarely prune nodes since, by definition, no intermediary primally valid solutions are ever found.

## 5.4 Comparison against "Learning to compare nodes"

Aside from comparisons against SCIP, we also compare against the previous state-of-the-art method by Labassi et al. (2022). One complication when benchmarking against Labassi et al. (2022) is that they assume

---

[5]We are not aware of a learned BnB node-selection heuristic used for MINLPs, so guidance towards feature selection doesn't exist yet. Taking advantage of them presumably also requires to train on nonlinear problems.

Table 2: Comparison against (Labassi et al., 2022). Note that we use a *single* network, evaluated out-of-distribution, while (Labassi et al., 2022) uses *different* networks, all trained on that specific type of mixed integer linear program. For all metrics, lower is better.

| | FMCNF | | GISP | | WPMS | |
|---|---|---|---|---|---|---|
| | Nodes | Runtime | Nodes | Runtime | Nodes | Runtime |
| Labassi FMCNF | $339.53 \pm 5.90$ | $29.45 \pm 2.13$ | — | — | — | — |
| Labassi GISP | — | — | $1219 \pm 1.73$ | $26.50 \pm 1.55$ | — | — |
| Labassi WPMS | — | — | — | — | $\mathbf{215.26 \pm 1.97}$ | $10.46 \pm 1.56$ |
| Ours | $\mathbf{187.33 \pm 3.67}$ | $\mathbf{19.99 \pm 2.25}$ | $\mathbf{1216.16 \pm 1.91}$ | $\mathbf{16.94 \pm 1.49}$ | $221.73 \pm 1.78$ | $\mathbf{8.04 \pm 1.72}$ |

a *separate model for each problem type*, while our method is already flexible enough to handle *all problem types within a single model*. Since Labassi et al. (2022) needs to re-train for each instance type and only works for linear problems, we cannot test it against MIPLIB and MINLPLIB. Instead, we reproduce all benchmarks used by Labassi et al. (2022) (each with their own separate models), and compare against them against our single model.

Labassi et al. (2022) considers the problem of *minimizing time-to-solution and minimizing the number of nodes explored* over a set of synthetic problems. This means that rather than the problem of minimizing the gap that can be reached within a given time limit (Table 1), here we always optimize to completion and track the runtime elapsed until the problem is solved. We use the "transfer" instances and best performing models provided by Labassi et al. (2022) and re-run the benchmarks on our hardware.

As we report in Table 2, we convincingly beat the prior work on every single benchmark (aside of WPMS node count), despite our method never having seen any of these problem types during training. This is surprising because Labassi et al. (2022) should have a convincing advantage due to the fact they not only have a dedicated agents for every single one of their problems, they also use the same generator for their training and testing instances. This implies that your single solver, evaluated out-of-distribution, manages to outperform the specialized agents proposed by Labassi et al. (2022) evaluated in-distribution. In general, our method improves upon Labassi et al. (2022) by between 30% and 56% with respect to runtime.

## 5.5 Ablations

We ablate the importance of the individual features used in our node selector using the SHAP (Lundberg & Lee, 2017) interpretability algorithm (see Appendix H). We find that our node selector automatically discovers best-practices from the classical node selection community, such as preferentially selecting nodes with lower lower-bound, balanced by greedy "best node" selection. Specifically interesting, we find that our selector significantly incorporates the amount of cutting planes applied to each node, hinting at a stronger than expected connection between node-utility and cutting planes.

In addition to the feature importance analysis done above, we also investigate running the model without a GNN: We find that when removing the GNN, the model tends to become very noisy and produce unreproducible experiments. Considering only the cases where the GNN-free model does well, we still find the model needs roughly 30% more nodes than the SCIP or our model with a GNN. More importantly, we notice the GNN-free model diverges during training: starting with a reward of roughly zero, the model diverges down to a reward of $\approx -0.2$, which amounts to a score roughly 20% worse than SCIP. We therefore conclude that, at least for our architecture, the GNN is necessary for both robustness and performance.

## 6 Limitations

There are still many open questions that give rise to future research or further investigation. First, while we report initial results in our ablation study (see Sec. 5.5 and Appendix H) on the importance of features, feature selection remains an area where we expect significant improvements, especially for nonlinear programming, which contemporary methods do not account for.

Second, one possible limitation is the limited runtime during training. Unfortunately, it is not computationally tractable for us to train on such large time-frames. However, we demonstrate in Sec. 5, that our model trained on a 45s time limit generalizes flawlessly to larger time limits of 5min. Hence we also expect our method to work on considerably larger problem sets.

Third, we also expect significant improvements in performance through code optimization. An important area for research lies in generalized instance generation: Instead of focusing on single domain instances for training (e.g. from TSP), an instance generator should create problem instances with consistent, but varying levels of difficulty across different problem domains. Further, the number of nodes used before switching to classical node selectors is only done heuristically. Finding optimal switching points between different node selectors is still an open problem even beyond learned solutions and represents an interesting place for further research.

## 7 Conclusion

We have proposed a novel approach to branch-and-bound node selection, that uses the structure of the branch-and-bound process and reinforcement learning to convincingly beat classical SCIP and learnt node selectors. By aligning our model with the branch-and-bound tree structure, we have demonstrated the potential to develop a versatile heuristic that can be applied across various optimization problem domains, despite being trained on a narrow set of instances. To our knowledge, this is the first demonstration of learned node selection to mixed-integer (nonlinear) programming.

### Acknowledgments

This work was supported by the Bavarian Ministry for Economic Affairs, Infrastructure, Transport and Technology through the Center for Analytics-Data-Applications (ADA-Center) within the framework of "BAYERN DIGITAL II".

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

## Appendix

## A    Features

Table 3 lists the features used on every individual node. The features split into two different types: One being "model" features, the other being "node" features. Model features describe the state of the entire model at the currently explored node, while node features are specific to the yet-to-be-solved added node. We aim to normalize all features with respect to problem size, as e. g., just giving the lower-bound to a problem is prone to numerical domain shifts. For instance a problem with objective $c^T x, x \in P$ is inherently the same from a solver point-of-view as a problem $10 c^T x, x \in P$, but would give different lower-bounds. Since NNs are generally nonlinear estimators, we need to make sure such changes do not induce huge distribution shifts. We also clamp the feature values between $[-10, 10]$ which represent "infinite" values, which can occur, for example in the optimality gap. Last but not least, we standardize features using empirical mean and standard deviation. These features are inspired by prior work, such as Labassi et al. (2022); Yilmaz & Yorke-Smith

Table 3: Features used per individual node.

| | | |
|---|---|---|
| model features | Number of cuts applied | normalized by total number of constraints |
| | Number of separation rounds | |
| | optimality gap | |
| | lp iterations | |
| | mean integrality gap | |
| | percentage of variables already integral | |
| | histogram of fractional part of variables | 10 evenly sized buckets |
| node features | depth of node | normalized by total number of nodes |
| | node lowerbound | normalized by min of primal and dual bound |
| | node estimate | normalized by min of primal and dual bound |

(2021), but adapted to the fact that we do not need e. g., explicit entries for the left or right child's optimality gap, as these (and more general K-step versions of these) can be handled by the GNN.

Further, to make batching tractable, we aim to have constant size features. This is different from e. g., Labassi et al. (2022), who utilize flexibly sized graphs to represent each node. The upside of this approach is that certain connections between variables and constraints may become more apparent, with the downside being the increased complexity of batching these structures and large amounts of nodes used. This isn't a problem for them, as they only consider pairwise comparisons between nodes, rather than the entire branch-and-bound graph, but for us would induce a great deal of complexity and computational overhead, especially in the larger instances. For this reason, we represent flexibly sized inputs, such as the values of variables, as histograms: i.e., instead of having $k$ nodes for $k$ variables and wiring them together, we produce once distribution of variable values with 10-buckets, and feed this into the network. This looses a bit of detail in the representation, but allows us to scale to much larger instances than ordinarily possible.

In general, these features are not optimized, and we would expect significant improvements from more well-tuned features. Extracting generally meaningful features from branch-and-bound is a nontrivial task and is left as a task for future work.

## B    Theoretical Derivation

One naive way of parameterizing actions is selecting probabilistically by training the probability of going to the left or right child at any node. This effectively gives a hierarchical Bernoulli description of finding a path in the tree. A path is simply a sequence of left ("zero direction") and right ("one direction") choices, with the total probability of observing a path being the likelihood of observing a specific chain

$$p(\text{leaf}) = \prod_{i \in P(\text{root}, \text{leaf})} p_i,$$

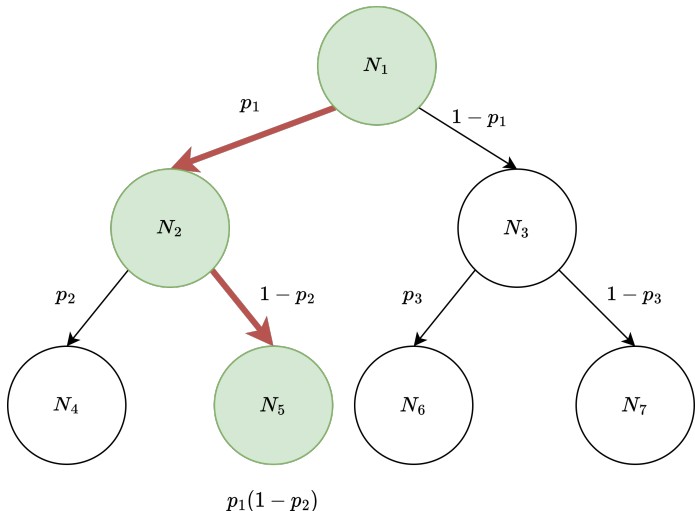

Figure 2: Naive approach using recursive selection. The probabilities are computed based on which "fork" of the tree is traveled. Sampling this can be done by sampling left or right based on $p_i$

with $p_i$ being the i'th decision in the graph (see Fig 2). Node selection can now be phrased as the likelihood of a random (weighted) walk landing at the specified node. However, using this parametrization directly would yield a significant number of problems:

Removing illegal paths from the model is quite challenging to do with sufficiently high performance. If a node gets pruned or fully solved, the probability to select this node has to be set to zero. This information has to be propagated upward the tree to make sure that the selection never reaches a "dead-end". This is possible, but turns out to be rather expensive to compute since it needs to evaluate the entire tree from leaves to root.

From a theoretical point of view, the hierarchical Bernoulli representation also has a strong prior towards making selections close to the root, as the probability of reaching a node a depth $K$ consists of $p_1 \cdot p_2 \cdots \cdot p_K$ selections. Since all $p_i$ are set to an uninformative prior $0 < p_i < 1$, the model at initialization has an increased likelihood to select higher nodes in the tree. Considering that classical methods specifically mix plunging heuristics into their selection to get increased depth exploration (see Sec. 3), this "breadth over depth" initialization is expected to give low performance. Therefore, one would need to find a proper initialization that allows the model to e. g., uniformly select nodes independent of depth to get good initialization.

The fact that multiple sampling sites exist in this method also poses issues with RL, as small errors early in the tree can have catastrophic effects later. This means the tree based model is a significantly higher variance estimator than the estimator we propose in Sec 4.3, which yields much worse optimization characteristics. It is well established in existing deep learning (e. g., (Kingma & Welling, 2013)) and general Bayesian Inference (e. g., (Robert & Roberts, 2021; Gelfand & Smith, 1990)) literature, that isolating or removing sampling sites can lead to lower variance, and therefore much more performant, statistical estimators.

Using those insights, we can now rewrite this naive approach into the one we propose in Sec. 4.3: First, instead of sampling just one path, we can sample *all* possible paths and compute the likelihood for each. The result will be a probability $p$ at every possible leaf. If one parameterizes this as a logarithmic probability the likelihood of sampling any individual leaf, can be written as

$$\log p_{\text{leaf}} = \sum_{i \in P(root, \text{leaf})} \log p_i. \tag{13}$$

We can further assume that the $\log p$'s are given as *unnormalized logits* $W'$ and only normalize them at the end such that to likelihood of all leaves together is 1.

This already gives an increase in performance, as the paths can be computed in parallel *without* needing iterated random sampling, which tends to be a very expensive operation. One also has fewer discrete choices,

meaning fewer backpropagation graph-breaks, which improves overall stability. The change to unnormalized logits acts essentially as a reparametrization trick (for other examples of reparametrization, see (Kingma & Welling, 2013)).

If one considers unnormalized log-probabilities, which have a range $(-\infty, \infty)$, this scheme becomes very similar to the one we ended up using in Sec. 4.3. Let $f(\text{node}, \text{tree})$ denote the unnormalized probabilities such that $p(\text{node}, \text{tree}) = \frac{f(\text{node}, \text{tree})}{Z}$. One can compute the likelihood of selecting a specific node as:

$$
\begin{aligned}
p(\text{leaf}|\text{tree}) &= \frac{f(\text{leaf}, \text{tree})}{\sum_{c \in \mathscr{C}} f(c, \text{tree})} && \text{normalization} \\
&= \frac{\exp(\log(f(\text{leaf}, \text{tree})))}{\sum_{c \in \mathscr{C}} \exp(\log(f(c, \text{tree})))} && \log f \text{ are unnormalized logits} \\
&= \frac{\exp(W'(\text{leaf}))}{\sum_{c \in \mathscr{C}} \exp(W'(c))} && \text{definition W' as unnormalized logit} \\
&= \text{softmax}_{\text{leaf}}\{W'(c)|\forall c \in \mathscr{C}\} && \text{definition softmax,}
\end{aligned}
$$

where $\text{softmax}_{\text{leaf}}$ refers to the softmax-entry associated with leaf. The resulting decomposition is, while not the same, equivalent to the original Bernoulli decomposition (in the sense that for any Bernoulli decomposition there exists a softmax-based decomposition). Beyond the massive reduction in sampling sites from $O(depth)$ to a single sampling site, phrasing the problem using unnormlized-$\log p$ representations gives rise to additional optimizations:

Firstly, to achieve uniform likelihood for all nodes at the beginning of training, we simply have to initialize $W(n) = 0$ for all nodes, which can be done by setting the output weights and biases to zero (in practice we set bias to zero and weights to a small random value).

Secondly to account for pruned or solved nodes, we do not need to change the unnormalized probabilities. Instead, we simply only sample the paths associated with selectable nodes, as we know the likelihood of selecting a pruned or solved node is zero. This means we can restrict ourselves to only evaluating the candidate paths $\mathscr{C}$, rather than all paths (see Eq. 9).

The last modification we made in our likelihood computation (see Eq. 8), is to normalize the weights based on depth. This is done mainly for numerical stability, but in our Bayesian framework is equivalent to computing the softmax-normalization Eq. 9 using weights inversely proportional to depth. The resulting algorithm is much more stable and performant than the naive parametrization. In early implementations we used the recursive-sampling approach to compute likelihoods, but the resulting scheme did not manage to reach above random results on the training set, presumably due to the higher computational burden and worse initialization. There are also other advantages to our parametrization, such as the fact that one can easily include a sampling temperatur $\tau$ into the process

$$
\text{node} \sim \text{softmax}\{W'(c)/\tau|\forall c \in \mathscr{C}\}
$$

which can be nice to tune the model more into the direction "exploitation" during testing. For our later benchmarking (Sec 5) we keep $\tau = 1$ (i.e., no temperature), as it would introduce a further tunable hyperparameter, but this could be interesting for future work.

This construction has interesting parallels to Monte-Carlo Tree Search (MCTS) Silver et al. (2017) based reinforcement learning methods. The difference in our method is that we consider iterative updates to a policy distribution $\pi$ rather than a Q-function, and therefore can be seen as a policy-based tree search scheme: While traditional MCTS uses variants of UCB-scores to guide the search, our method can be thought of as variants of Thompson sampling to facilitate the exploration-exploitation tradeoff. We leave a more thorough investigation of this to future work.

Table 4: Performance across benchmarks (the policy only saw TSP instances during training). The 5min runs use the same model, evaluated for the first 650 nodes, and processed according to Sec. 5.1.

| Benchmark | Reward | Utility | Utility/Node | Win-rate | geo-mean Ours | geo-mean SCIP |
|---|---|---|---|---|---|---|
| TSPLIB (Reinelt, 1991) | 0.184 | 0.030 | 0.193 | 0.50 | 0.931 | 0.957 |
| UFLP (Kochetov & Ivanenko, 2005) | 0.078 | 0.093 | -0.064 | 0.636 | 0.491 | 0.520 |
| MINLPLib (Bussieck et al., 2003) | 0.487 | 0.000 | 0.114 | 0.852 | 28.783 | 31.185 |
| MIPLIB (Gleixner et al., 2021) | 0.140 | -0.013 | 0.208 | 0.769 | 545.879 | 848.628 |
| TSPLIB@5min | 0.192 | 0.056 | 0.336 | 0.600 | 1.615 | 2.000 |
| MINLPlib@5min | 0.486 | -0.012 | 0.078 | 0.840 | 17.409 | 20.460 |
| MIPlib@5min | 0.150 | -0.075 | 0.113 | 0.671 | 66.861 | 106.400 |

## C    Additional metrics

Aside from the previously discussed Reward in Section 4.1, we also considered two different ways of measuring the quality of our method compared to SCIP. While both of these have theoretical problems, we still find them interesting to consider as they act as additional reference points for the behavior of the policies.

**Utility** defines the *difference* between both methods normalized using the maximum of both gaps:

$$\text{Utility} = \left( \frac{gap(scip) - gap(node\ selector)}{\max\left(gap(node\ selector), gap(scip)\right)} \right). \tag{14}$$

The reason we do not use this as a reward measure is because we empirically found it to produce worse models. This is presumably because some of the negative attributes of our reward, e.g., the asymmetry of the reward signal, lead to more robust policies. In addition, the utility metric gives erroneous measurements when both models hit zero optimality gap. This is because utility implicitly defines $\frac{0}{0} = 0$, rather than reward, which defines it as $\frac{0}{0} = 1$. In some sense the utility measurement is accurate, in that our method does not improve upon the baseline. On the other hand, our method is already provably optimal as soon as it reaches a gap of 0%. In general, utility compresses the differences more than reward which may or may not be beneficial in practice.

**Utility per Node** normalizes Utility by the number of nodes used during exploration:

$$\text{Utility/Node} = \left( \frac{scip - selector}{\max\left(selector, scip\right)} \right), \tag{15a}$$

where $selector = \frac{gap(node\ selector)}{nodes(node\ selector)}$ and $scip = \frac{gap(scip)}{nodes(scip)}$. The per-node utility gives a proxy for the total amount of "work" done by each method. However, it ignores the individual node costs, as solving the different LPs may take different amounts of resources (a model with higher "utility/node" is not necessarily more efficient as our learner might pick cheap but lower expected utility nodes on purpose). Further, the metric is underdefined: comparing two ratios, a method may become better by increasing the number of nodes processed, but keeping the achieved gap constant. In practice the number of nodes processed by our node selector is dominated by the implementation rather than the node choices, meaning we can assume it is invariant to changes in policy. Another downside arises if both methods reach zero optimality gap, the resulting efficiency will also be zero regardless of how many nodes we processed. As our method tend to reach optimality much faster (see Sec. 5 and Appendix I), all utility/node results can be seen as a lower-bound for the actual efficiency.

## D    Comparisons of Runtime to completion

We also compare the runtime of our selector against the SCIP baseline on TSPLIB. For this we set a maximum time limit of 30min and run both methods. It is worth noting that due to the inherent difficulty of TSPLIB, a significant number of problems are still unsolved after 30min, in which case these models are simply assigned the maximum time as their overall completion time. Note that this also implies that the SCIP solver itself not capable of solving these problems due to their scale. We reckon that this is also the reason why e.g., Labassi

| Method | Runtime shifted geometric mean | Nodes Processed geometric mean |
|--------|-------------------------------|-------------------------------|
| SCIP   | 1016.61s                      | 4122.67                       |
| Ours   | 1003.03s                      | 2735.33                       |

Table 5: The shifted geometric mean (shifted by +10) of runtime and number of nodes processed.

et al. (2022) does not even attempt to benchmark on these large scale problems. Our method outperforms the baseline, however due to the fact that our method aims to improve the quality of early selections in particular, the amount of improvement is rather small. Further, our method is fundamentally undertrained to adequately deal with longer time horizons and we would assume the effect of our method is larger the longer the model is trained. Despite this being a worst-case evaluation scenario, our method manages to outperform the baseline in both runtime and number of nodes needed.

Interestingly, we also manage to reach this superior performance while utilizing only about half the number of nodes. This discrepancy cannot be explained by just considering the slowdown due to our policy needing to be evaluated, as, while it is a significant overhead due to the Python implementation, it is very far from being large enough to justify a 50% slowdown over the course of 30min. This also shows more generally that even if one restricts themselves to only processing the beginning of the branch-and-bound tree, one can reach superior performance to the standard node-selection schemes.

# E  Architecture

Our network consists of two subsystems: First, we have the feature embedder that transforms the raw features into embeddings, without considering other nodes this network consists of one linear layer $|d_{features}| \rightarrow |d_{model}|$ with LeakyReLU (Xu et al., 2015) activation followed by two $|d_{model}| \rightarrow |d_{model}|$ linear layers (activated by LeakyReLU) with skip connections. We finally normalize the outputs using a Layernorm (Ba et al., 2016) *without* trainable parameters (i. e., just shifting and scaling the feature dimension to a normal distribution).

Second, we consider the GNN model, whose objective is the aggregation across nodes according to the tree topology. This consists of a single LeakyReLU activated layer with skip-connections. We use ReZero (Bachlechner et al., 2020) initialization to improve the convergence properties of the network. Both the weight and value heads are simple linear projections from the embedding space. Following the guidance in (Andrychowicz et al., 2020), we make sure the value and weight networks are independent by detaching the value head's gradient from the embedding network.

For this work we choose $|d_{model}| = 512$, but we did not find significant differences between different model sizes past a width of 256. For training we use AdamW Loshchilov & Hutter (2017) with a standard learning rate of $3 \cdot 10^{-4}$ and default PPO parameters.

# F  TSP-as-MILP Formulation

In general, due to the fact that TSP is amongst the most studied problems in discrete optimization, we can expect existing mixed-integer programming systems to have rich heuristics that provide a strong baseline for

our method. Mathematically, we choose the Miller–Tucker–Zemlin (MTZ) formulation (Miller et al., 1960):

$$\min_x \quad \sum_{i=1}^{n} \sum_{j \neq i, j=1}^{n} c_{ij} x_{ij}$$

$$\text{subject to} \quad \sum_{j=1, i \neq j}^{n} x_{ij} = 1 \qquad \forall i = 1, \ldots, n$$

$$\sum_{i=1, i \neq j}^{n} x_{ij} = 1 \qquad \forall j = 1, \ldots, n$$

$$u_1 - u_j + (n-1)x_{ij} \leq n - 2 \quad 2 \leq i \neq j \leq n$$

$$2 \leq u_i \leq n \qquad 2 \leq i \leq n$$

$$u_i \in \mathbb{Z}, x_{ij} \in \{0, 1\}$$

Effectively this formulation keeps two buffers: one being the actual $(i, j)$-edges travelled $x_{ij}$, the other being a node-order variable $u_i$ that makes sure that $u_i < u_j$ if $i$ is visited before $j$. There are alternative formulations, such as the Dantzig–Fulkerson–Johnson (DFJ) formulation, which are used in modern purpose-built TSP solvers, but those are less useful for general problem generation: The MTZ formulation essentially relaxes the edge-assignments and order constraints, which then are branch-and-bounded into hard assignments during the solving process. This is different to DFJ, which instead relaxes the "has to pass through all nodes" constraint. DFJ allows for subtours (e. g., only contain node $A, B, C$ but not $D, E$) which then get slowly eliminated via the on-the-fly generation of additional constraints. To generate these constraints one needs specialised row-generators which, while very powerful from an optimization point-of-view, make the algorithm less general as a custom row-generator has to intervene into every single node. However, in our usecase we also do not really care about the ultimate performance of individual algorithms as the reinforcement learner only looks for improvements to the existing node selections. This means that as long as the degree of improvement can be adequately judged, we do not need state-of-the-art solver implementations to give the learner a meaningful improvement signal.

## G  Uncapacitated facility location Problem

Mathematically, the uncapacitated facility location problem can be seen as sending a product $z_{ij}$ from facility $i$ to consumer $j$ with cost $c_{ij}$ and demand $d_j$. One can only send from $i$ to $j$ if facility $i$ was built in the first place, which incurs cost $f_i$. The overall problem therefore is

$$\min_x \quad \sum_{i=1}^{n} \sum_{i=1}^{m} c_{ij} d_j z_{ij} + \sum_{i=0}^{n} f_i x_i$$

$$\text{subject to} \quad \sum_{j=1, i \neq j}^{n} z_{ij} = 1 \qquad \forall i = 1, \ldots, m$$

$$\sum_{i=1, i \neq j}^{n} z_{ij} \leq M x_i \qquad \forall j = 1, \ldots, n$$

$$z_{ij} \in \{0, 1\} \qquad \forall i, j = 1, \ldots, n$$

$$x_i \in \{0, 1\} \qquad \forall i = 1, \ldots, n$$

where $M$ is a suitably large constant representing the infinite-capacity one has when constructing $x_i = 1$. One can always choose $M \geq m$ since that, for the purposes of the polytop is equivalent to setting $M$ to literal infinity. This is also sometimes referred to as the "big $M$" method.

The instance generator by Kochetov & Ivanenko (2005) works by setting $n = m = 100$ and setting all opening costs at 3000. Every city has 10 "cheap" connections sampled from $\{0, 1, 2, 3, 4\}$ and the rest have cost 3000, which represents infinity (i. e., also invoking the big $M$ method).

# H   Feature Importance

We try to interpret the learned node-selection heuristic by applying the KernelSHAP (Lundberg & Lee, 2017) method. SHAP methods try to estimate the feature importance by measuring the performance of an estimator where some portion of the features are replaced with a "neutral" element. Doing this with a large enough set, one is able to extract the feature importance in the form of the magnitude of the change between the expected value of the estimator and the value the estimator would have had if a specific feature were absent.

Analyzing our RL method with SHAP has two significant limitations: Firstly, we ignore the impact of message passing on the model and instead analyze every node as a leaf node. This is of relatively small impact as the time where a node was selected it had to have been a leaf node anyways, so the plots faithfully show the model output at selection time.

The second limitation is that we do not have an iid. dataset, but instead have to build one from the environment first. To build that dataset, we could sample random BnB trees and evaluate on the nodes, but this would mean we evaluate our model completely out-of-distribution, as random selection will most likely lead to states the model would never want to reach and is therefore ill equipped to handle. Therefore, we instead build our dataset by running our node-selector on a subset of MIPLIB to gather our importance-evaluation data. This has the downside that the expected value of our selection network is higher than in random selection, since the model is guided towards selecting higher value nodes.

With these caveats in mind, one can see in fig. 3 that our node selector relies significantly on the node lowerbound for setting its weight. This makes sense: in node-selection there is a fundamental tension between raising the lower bound and opportunistically searching for new primal solutions. The current SCIP default "hybrid best bound search" does exactly that, just using hand-made heuristics where a nodes are selected based on a mixture of following the currently best bound nodes, and plunging depth-first search to find better solutions. However, as can be seen in our benchmarks (tab 1), we beat SCIP's default selector quite significantly, which implies a nontrivial interaction in the remaining features.

Looking at the beeswarm plot 4, we observe that our model preferentially picks nodes which, in addition to a low node lowerbound, also uses nodes which have few variables with an integrality gap of 0.2 and 0.3, has a lot of cuts applied, and has a high expected integrality gap. If we considered a node with all of these features (and assume there would not be any nonlinear value-dampening effects), such a node would be a prime candidate for having good child nodes: A node with a low lowerbound means that the node still has significant room for improvement, while having a lot of cuts applied means that the node's solution is sufficiently close to existing integral points to make it feasible to find better values. Looking further into the features, one can also see that the model dislikes nodes that already have a significant number of "already integral" variables. This makes sense as nodes with a significant number of integral variables that are not solved already (i. e., they have children to split on), can be quite hard to complete while not giving a significant improvement in solution quality. This is because most likely a primal heuristic has already found the optimal value of that node, but branch-and-bound still needs a significant number of trials to prove that the found value is optimal in that subtree.

In short, our solver picks the nodes with the highest degree of possible improvement (low lowerbound), while also favoring nodes that have more information (more cuts applied) and presumably are nontrivial to solve via heuristics (high mean to integral gap/low "already integral" ratio).

In general, interpreting dynamic RL policies is highly nontrivial, especially if the estimator is nonlinear, but the interactions we can clearly see in the SHAP plots indicates that the learned policy is reasonably well grounded in existing best-practices for node selectors.

# I   Full Results

The following two sections contain the per-instance results on the two "named" benchmarks TSPLIB (Reinelt, 1991) and MINLPLIB (Bussieck et al., 2003). We test against the strong SCIP 8.0.4 baseline. Due to compatibility issues, we decided not to test against (Labassi et al., 2022) or (He et al., 2014): These

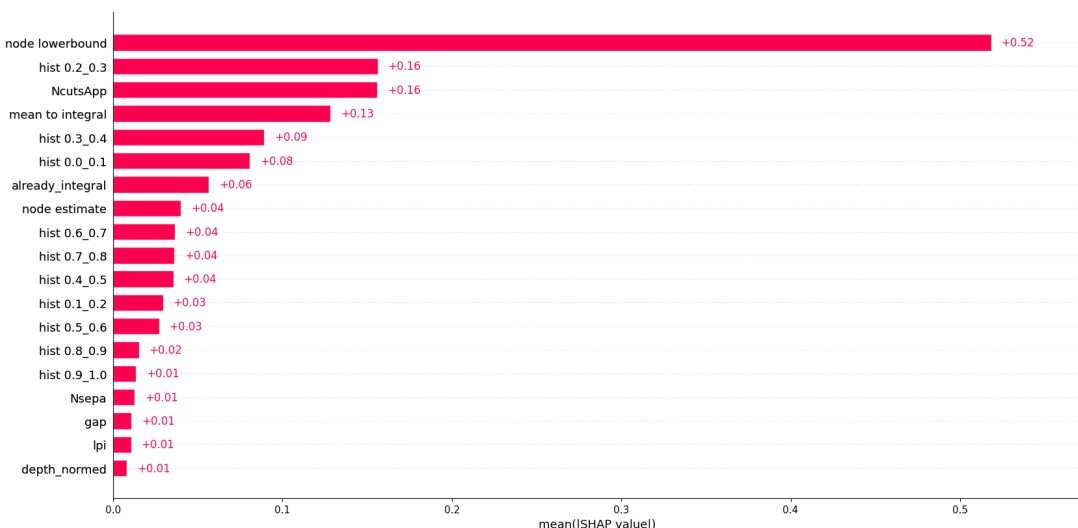

Figure 3: Barplot showing feature importance of individual nodes using the KernelSHAP (Lundberg & Lee, 2017) method.

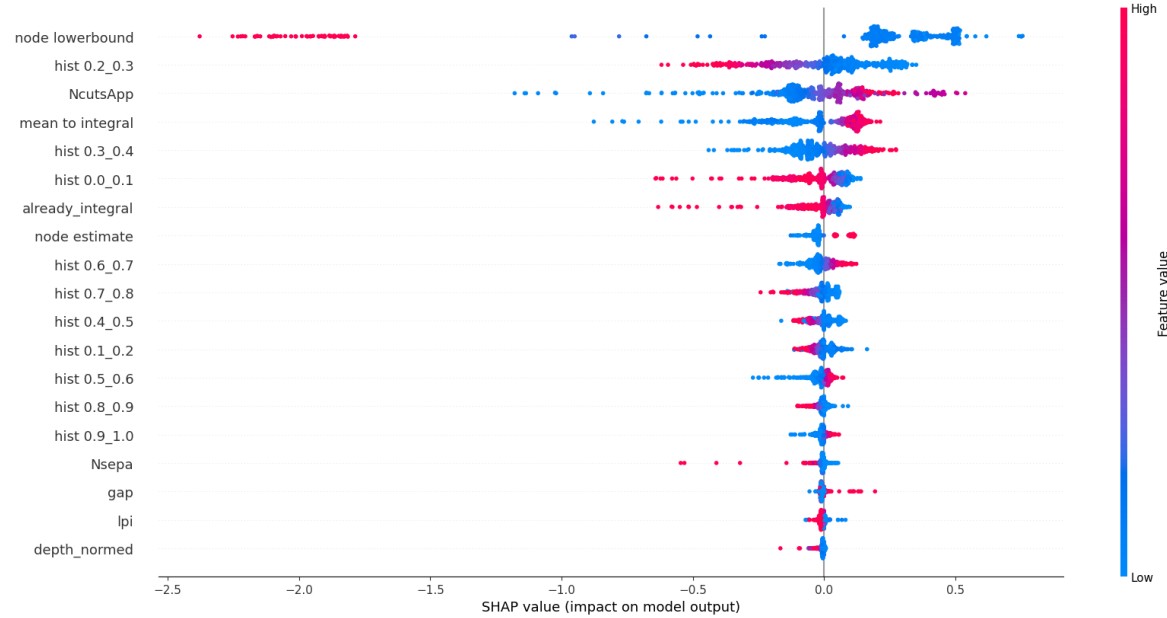

Figure 4: Beeswarm plot showing feature importance of individual nodes using the KernelSHAP (Lundberg & Lee, 2017) method.

methods were trained against older versions of SCIP, which not only made running them challenging, but also would not give valid comparisons as we cannot properly account for changes between SCIP versions. Labassi et al. (2022) specifically relies on changes to the SCIP interface, which makes porting to SCIP 8.0.4 intractable. In general, this shouldn't matter too much, as SCIP is still demonstrably the state-of-the-art non-commercial mixed-integer solver, which frequently outperforms even closed-source commercial solvers (see Mittelmann (2021) for thorough benchmarks against other solvers), meaning outperforming SCIP can be seen as outperforming the state-of-the-art.

### I.1 Kochetov-UFLP

To demonstrate the generalizability of the learned heuristics, we test our method on the Uncapacitated Facility Location Problem (see Appendix G) *without further finetuning*, i.e., we only train on TSP instances and never show the algorithm any other linear or nonlinear problem. For testing, we generate 1000 instances using the well-known problem generator by Kochetov & Ivanenko (2005), which was designed to have large optimality gaps, making these problems particularly challenging.

Our method performs very similar to the highly optimized baseline, despite never having seen the UFL problem, see Table 1. We argue that this is specifically because our method relies on tree-wide behaviour, rather than individual features to make decisions. We further hypothesize that the reason for the advantage over the baseline being so small is due to the fact that UFLP consists of "adversarial examples" to the branch-and-bound method where cuts have reduced effectiveness. This means clever node-selection strategies have limited impact on overall performance.

An interesting aspect is that our method processes more nodes than the baseline, which also leads to the loss in node-efficiency. This implies that our method selects significantly easier nodes, as ordinarily our solver is slower just due to the additional overhead. Considering that this benchmark was specifically designed to produce high optimality gaps, it makes sense that our solver favours node quantity over quality, which is an interesting emergent behaviour of our solver.

### I.2 TSPLIB results

Table 6: Results on TSPLIB (Reinelt, 1991) after 45s runtime. Note that we filter out problems in which less than 5 nodes were explored as those problems cannot gain meaningful advantages even with perfect node selection. "Name" refers to the instances name, "Gap Base/Ours" corresponds to the optimization gap achieved by the baseline and our method respectively (lower is better), "Nodes Base/Ours" to the number of explored Nodes by each method, and "Reward", "Utility" and "Utility Node" to the different performance measures as described in Section 5.

| Name | Gap Ours | Gap Base | Nodes Ours | Nodes Base | Reward | Utility | Utility/Node |
|---|---|---|---|---|---|---|---|
| att48 | 0.287 | 0.286 | 1086 | 2670 | -0.002 | -0.002 | 0.593 |
| bayg29 | 0.000 | 0.000 | 2317 | 7201 | 1.000 | 0.000 | 0.000 |
| bays29 | 0.000 | 0.036 | 11351 | 10150 | 1.000 | 1.000 | 0.997 |
| berlin52 | 0.000 | 0.000 | 777 | 1634 | 1.000 | 0.000 | 0.000 |
| bier127 | 2.795 | 2.777 | 23 | 25 | -0.007 | -0.007 | 0.074 |
| brazil58 | 0.328 | 0.644 | 1432 | 2182 | 0.491 | 0.491 | 0.666 |
| burma14 | 0.000 | 0.000 | 96 | 65 | 1.000 | 0.000 | 0.000 |
| ch130 | 8.801 | 8.783 | 48 | 43 | -0.002 | -0.002 | -0.106 |
| ch150 | 7.803 | 7.802 | 18 | 18 | -0.000 | -0.000 | -0.000 |
| d198 | 0.582 | 0.582 | 10 | 11 | -0.000 | -0.000 | 0.091 |
| dantzig42 | 0.185 | 0.100 | 2498 | 3469 | -0.847 | -0.459 | -0.248 |
| eil101 | 2.434 | 2.430 | 31 | 61 | -0.002 | -0.002 | 0.491 |
| eil51 | 0.178 | 0.017 | 828 | 4306 | -1.000 | -0.907 | -0.514 |
| eil76 | 0.432 | 1.099 | 309 | 709 | 0.607 | 0.607 | 0.829 |
| fri26 | 0.000 | 0.000 | 1470 | 6721 | 1.000 | 0.000 | 0.000 |
| gr120 | 7.078 | 7.083 | 41 | 43 | 0.001 | 0.001 | 0.047 |
| gr137 | 0.606 | 0.603 | 30 | 25 | -0.006 | -0.006 | -0.171 |
| gr17 | 0.000 | 0.000 | 92 | 123 | 1.000 | 0.000 | 0.000 |
| gr24 | 0.000 | 0.000 | 110 | 207 | 1.000 | 0.000 | 0.000 |
| gr48 | 0.192 | 0.340 | 586 | 2479 | 0.435 | 0.435 | 0.866 |
| gr96 | 0.569 | 0.552 | 93 | 182 | -0.032 | -0.031 | 0.472 |
| hk48 | 0.071 | 0.106 | 2571 | 2990 | 0.324 | 0.324 | 0.419 |
| kroA100 | 8.937 | 8.945 | 102 | 233 | 0.001 | 0.001 | 0.563 |
| kroA150 | 11.343 | 11.340 | 23 | 21 | -0.000 | -0.000 | -0.087 |
| kroA200 | 13.726 | 13.723 | 5 | 7 | -0.000 | -0.000 | 0.286 |
| kroB100 | 7.164 | 7.082 | 83 | 109 | -0.011 | -0.011 | 0.230 |
| kroB150 | 10.965 | 10.965 | 16 | 14 | 0.000 | 0.000 | -0.125 |
| kroB200 | 11.740 | 11.740 | 7 | 6 | 0.000 | 0.000 | -0.143 |
| kroC100 | 8.721 | 8.754 | 118 | 133 | 0.004 | 0.004 | 0.116 |
| kroD100 | 7.959 | 7.938 | 70 | 111 | -0.003 | -0.003 | 0.368 |
| kroE100 | 8.573 | 2.952 | 105 | 108 | -1.000 | -0.656 | -0.646 |
| lin105 | 2.005 | 2.003 | 98 | 149 | -0.001 | -0.001 | 0.341 |
| pr107 | 1.367 | 1.336 | 128 | 217 | -0.024 | -0.023 | 0.396 |
| pr124 | 0.937 | 0.935 | 64 | 61 | -0.001 | -0.001 | -0.048 |
| pr136 | 2.351 | 2.350 | 31 | 45 | -0.000 | -0.000 | 0.311 |
| pr144 | 2.228 | 2.200 | 47 | 37 | -0.012 | -0.012 | -0.222 |
| pr152 | 2.688 | 2.683 | 14 | 41 | -0.002 | -0.002 | 0.658 |
| pr226 | 1.091 | 1.092 | 6 | 6 | 0.001 | 0.001 | 0.001 |
| pr76 | 0.534 | 0.476 | 201 | 855 | -0.123 | -0.109 | 0.736 |
| rat99 | 0.853 | 0.849 | 41 | 80 | -0.005 | -0.005 | 0.485 |
| rd100 | 5.948 | 4.462 | 100 | 166 | -0.333 | -0.250 | 0.197 |
| si175 | 0.270 | 0.270 | 8 | 7 | 0.000 | 0.000 | -0.125 |
| st70 | 0.586 | 3.018 | 379 | 1068 | 0.806 | 0.806 | 0.931 |
| swiss42 | 0.000 | 0.000 | 1075 | 1133 | 1.000 | 0.000 | 0.000 |
| ulysses16 | 0.000 | 0.000 | 18322 | 19553 | 1.000 | 0.000 | 0.000 |
| ulysses22 | 0.103 | 0.127 | 13911 | 13313 | 0.191 | 0.191 | 0.154 |
| Mean | — | — | 1321 | 1799 | **0.184** | **0.030** | **0.193** |

## I.3 MIPLIB results

Table 7: Results on MIPLIB (Gleixner et al., 2021) after 45s runtime. Note that we filter out problems in which less than 5 nodes were explored as those problems cannot gain meaningful advantages even with perfect node selection. "Name" refers to the instances name, "Gap Base/Ours" corresponds to the optimization gap achieved by the baseline and our method respectively (lower is better), "Nodes Base/Ours" to the number of explored Nodes by each method, and "Reward", "Utility" and "Utility Node" to the different performance measures as described in Section 5. Note that all results where achieved with a policy only trained on TSP instances

| Name | Gap Ours | Gap Base | Nodes Ours | Nodes Base | Reward | Utility | Utility/Node |
|---|---|---|---|---|---|---|---|
| 30n20b8 | 2.662 | ∞ | 147 | 301 | 1.000 | 1.000 | 1.000 |
| 50v-10 | 0.101 | 0.113 | 303 | 1094 | 0.103 | 0.103 | 0.752 |
| CMS750_4 | 0.100 | 0.072 | 68 | 281 | -0.389 | -0.280 | 0.664 |
| air05 | 0.000 | 0.000 | 248 | 523 | 1.000 | 0.000 | 0.000 |
| assign1-5-8 | 0.085 | 0.087 | 17466 | 23589 | 0.030 | 0.030 | 0.282 |
| binkar10_1 | 0.000 | 0.000 | 2843 | 2270 | 1.000 | 0.000 | 0.000 |
| blp-ic98 | 0.127 | 0.127 | 26 | 43 | 0.001 | 0.001 | 0.396 |
| bnatt400 | ∞ | ∞ | 547 | 1568 | 0.000 | 0.000 | 0.651 |
| bnatt500 | ∞ | ∞ | 148 | 936 | 0.000 | 0.000 | 0.842 |
| bppc4-08 | 0.038 | 0.038 | 1318 | 3277 | 0.000 | 0.000 | 0.598 |
| cost266-UUE | 0.130 | 0.143 | 468 | 770 | 0.094 | 0.094 | 0.449 |
| csched007 | ∞ | ∞ | 558 | 1770 | 0.000 | 0.000 | 0.685 |
| csched008 | 0.070 | ∞ | 910 | 1179 | 1.000 | 1.000 | 1.000 |
| cvs16r128-89 | 0.560 | 0.601 | 6 | 7 | 0.068 | 0.068 | 0.202 |
| drayage-25-23 | 0.000 | 0.000 | 105 | 267 | 1.000 | 0.000 | 0.000 |
| dws008-01 | ∞ | ∞ | 123 | 173 | 0.000 | 0.000 | 0.289 |
| eil33-2 | 0.194 | 0.189 | 191 | 171 | -0.025 | -0.024 | -0.127 |
| fast0507 | 0.027 | 0.027 | 11 | 7 | -0.003 | -0.003 | -0.366 |
| fastxgemm-n2r6s0t2 | 18.519 | 18.519 | 785 | 2531 | 0.000 | 0.000 | 0.690 |
| fhnw-binpack4-4 | ∞ | ∞ | 140002 | 152608 | 0.000 | 0.000 | 0.083 |
| fhnw-binpack4-48 | ∞ | 0.000 | 15019 | 24649 | -1.000 | -1.000 | -1.000 |
| fiball | 0.029 | 0.036 | 442 | 610 | 0.200 | 0.200 | 0.420 |
| gen-ip002 | 0.008 | 0.010 | 88794 | 125319 | 0.197 | 0.197 | 0.397 |
| gen-ip054 | 0.008 | 0.010 | 157950 | 179874 | 0.207 | 0.207 | 0.263 |
| glass-sc | 0.580 | 0.495 | 200 | 328 | -0.173 | -0.148 | 0.285 |
| glass4 | 1.123 | 1.033 | 37424 | 35671 | -0.087 | -0.080 | -0.123 |
| gmu-35-40 | 0.001 | 0.001 | 28534 | 27077 | 0.402 | 0.398 | 0.276 |
| gmu-35-50 | 0.001 | 0.001 | 16456 | 22333 | 0.177 | 0.176 | 0.346 |
| graph20-20-1rand | 0.000 | 0.000 | 416 | 283 | 1.000 | 0.000 | 0.000 |
| graphdraw-domain | 0.421 | 0.430 | 49640 | 56798 | 0.022 | 0.022 | 0.145 |
| ic97_potential | 0.023 | 0.040 | 39316 | 30633 | 0.415 | 0.415 | 0.247 |
| icir97_tension | 0.011 | 0.006 | 6697 | 7943 | -0.882 | -0.468 | -0.367 |
| irp | 0.000 | 0.000 | 6 | 6 | 1.000 | 0.000 | 0.000 |
| istanbul-no-cutoff | 0.514 | 0.393 | 37 | 28 | -0.309 | -0.236 | -0.422 |
| lectsched-5-obj | ∞ | 2.200 | 1192 | 1118 | -1.000 | -1.000 | -1.000 |
| leo1 | 0.118 | 0.113 | 34 | 108 | -0.049 | -0.046 | 0.670 |
| leo2 | 0.345 | 0.135 | 49 | 61 | -1.000 | -0.609 | -0.514 |
| mad | ∞ | ∞ | 78783 | 81277 | 0.000 | 0.000 | 0.031 |
| markshare2 | ∞ | ∞ | 91135 | 127265 | 0.000 | 0.000 | 0.284 |
| markshare_4_0 | ∞ | ∞ | 570277 | 682069 | 0.000 | 0.000 | 0.164 |
| mas74 | 0.079 | 0.084 | 32005 | 26180 | 0.060 | 0.060 | -0.129 |
| mas76 | 0.014 | 0.015 | 49987 | 52401 | 0.060 | 0.060 | 0.100 |
| mc11 | 0.008 | 0.009 | 333 | 1989 | 0.139 | 0.138 | 0.855 |
| mcsched | 0.090 | 0.086 | 439 | 1526 | -0.049 | -0.046 | 0.698 |
| mik-250-20-75-4 | 0.000 | 0.000 | 10067 | 10120 | 1.000 | 0.000 | 0.000 |
| milo-v12-6-r2-40-1 | 0.038 | 0.031 | 340 | 514 | -0.242 | -0.195 | 0.179 |
| momentum1 | 2.868 | 2.868 | 10 | 9 | -0.000 | -0.000 | -0.100 |
| n2seq36q | 0.665 | 0.665 | 5 | 6 | 0.000 | 0.000 | 0.167 |
| n5-3 | 0.046 | 0.000 | 427 | 595 | -1.000 | -1.000 | -1.000 |
| neos-1171737 | 0.032 | 0.032 | 7 | 13 | 0.000 | 0.000 | 0.462 |
| neos-1445765 | 0.000 | 0.000 | 190 | 263 | 1.000 | 0.000 | 0.000 |
| neos-1456979 | ∞ | 0.344 | 204 | 405 | -1.000 | -1.000 | -1.000 |
| neos-1582420 | 0.016 | 0.016 | 11 | 11 | 0.000 | 0.000 | 0.000 |
| neos-2657525-crna | ∞ | ∞ | 42826 | 45188 | 0.000 | 0.000 | 0.052 |
| neos-2978193-inde | 0.013 | 0.013 | 964 | 2178 | 0.000 | 0.000 | 0.557 |
| neos-3004026-krka | ∞ | ∞ | 1134 | 1163 | 0.000 | 0.000 | 0.025 |
| neos-3024952-loue | ∞ | ∞ | 246 | 377 | 0.000 | 0.000 | 0.347 |
| neos-3046615-murg | 2.515 | 2.631 | 66921 | 79117 | 0.044 | 0.044 | 0.191 |
| neos-3083819-nubu | 0.000 | 0.000 | 1683 | 1687 | 1.000 | 0.000 | 0.000 |
| neos-3381206-awhea | 0.000 | 0.000 | 969 | 230 | 1.000 | 0.000 | 0.000 |
| neos-3402294-bobin | ∞ | ∞ | 10 | 24 | 0.000 | 0.000 | 0.583 |
| neos-3627168-kasai | 0.003 | 0.008 | 6269 | 3338 | 0.577 | 0.577 | 0.205 |

| Name | Gap Ours | Gap Base | Nodes Ours | Nodes Base | Reward | Utility | Utility/Node |
|---|---|---|---|---|---|---|---|
| neos-3754480-nidda | ∞ | ∞ | 87703 | 106632 | 0.000 | 0.000 | 0.178 |
| neos-4338804-snowy | 0.024 | 0.028 | 37447 | 36741 | 0.125 | 0.125 | 0.107 |
| neos-4387871-tavua | 0.631 | 0.634 | 5 | 7 | 0.005 | 0.005 | 0.289 |
| neos-4738912-atrato | 0.016 | 0.006 | 529 | 1064 | -1.000 | -0.634 | -0.265 |
| neos-4954672-berkel | 0.265 | 0.254 | 454 | 775 | -0.043 | -0.041 | 0.389 |
| neos-5093327-huahum | 0.539 | 0.559 | 5 | 6 | 0.036 | 0.036 | 0.197 |
| neos-5107597-kakapo | 2.639 | 5.077 | 1885 | 2332 | 0.480 | 0.480 | 0.580 |
| neos-5188808-nattai | ∞ | ∞ | 16 | 105 | 0.000 | 0.000 | 0.848 |
| neos-5195221-niemur | 106.417 | 106.417 | 11 | 12 | 0.000 | 0.000 | 0.083 |
| neos-911970 | 0.000 | 0.000 | 3905 | 15109 | 1.000 | 0.000 | 0.000 |
| neos17 | 0.000 | 0.000 | 2151 | 3346 | 1.000 | 0.000 | 0.000 |
| neos5 | 0.062 | 0.059 | 66231 | 91449 | -0.053 | -0.050 | 0.235 |
| neos859080 | 0.000 | 0.000 | 990 | 1227 | 1.000 | 0.000 | 0.000 |
| net12 | 2.592 | 2.114 | 56 | 29 | -0.227 | -0.185 | -0.578 |
| ns1208400 | ∞ | ∞ | 82 | 150 | 0.000 | 0.000 | 0.453 |
| ns1830653 | 2.831 | 1.242 | 334 | 686 | -1.000 | -0.561 | -0.099 |
| ns1952667 | ∞ | ∞ | 100 | 52 | 0.000 | 0.000 | -0.480 |
| nu25-pr12 | 0.000 | 0.000 | 119 | 153 | 1.000 | 0.000 | 0.000 |
| nursesched-sprint02 | 0.000 | 0.000 | 9 | 7 | 1.000 | 0.000 | 0.000 |
| nw04 | 0.000 | 0.000 | 6 | 6 | 1.000 | 0.000 | 0.000 |
| pg | 0.000 | 0.000 | 460 | 491 | 1.000 | 0.000 | 0.000 |
| pg5_34 | 0.004 | 0.004 | 275 | 592 | -0.023 | -0.022 | 0.524 |
| piperout-08 | 0.000 | 0.000 | 223 | 309 | 1.000 | 0.000 | 0.000 |
| piperout-27 | 0.000 | 0.000 | 47 | 28 | 1.000 | 0.000 | 0.000 |
| pk1 | 1.244 | 1.117 | 102268 | 120685 | -0.113 | -0.102 | 0.057 |
| radiationm18-12-05 | 0.057 | 0.167 | 886 | 2569 | 0.661 | 0.661 | 0.883 |
| rail507 | 0.033 | 0.033 | 10 | 9 | 0.000 | 0.000 | -0.100 |
| ran14x18-disj-8 | 0.115 | 0.092 | 458 | 975 | -0.251 | -0.200 | 0.412 |
| rd-rplusc-21 | ∞ | ∞ | 137 | 3542 | 0.000 | 0.000 | 0.961 |
| reblock115 | 0.106 | 0.139 | 80 | 731 | 0.238 | 0.238 | 0.917 |
| rmatr100-p10 | 0.216 | 0.326 | 43 | 74 | 0.337 | 0.337 | 0.615 |
| rocI-4-11 | 0.671 | 0.837 | 12054 | 7909 | 0.198 | 0.198 | -0.181 |
| rocII-5-11 | 3.479 | 1.568 | 164 | 287 | -1.000 | -0.549 | -0.211 |
| rococoB10-011000 | 1.244 | 1.258 | 12 | 26 | 0.012 | 0.012 | 0.544 |
| rococoC10-001000 | 0.337 | 0.153 | 135 | 866 | -1.000 | -0.546 | 0.656 |
| roll3000 | 0.000 | 0.000 | 1156 | 2046 | 1.000 | 0.000 | 0.000 |
| sct2 | 0.001 | 0.002 | 2117 | 1215 | 0.619 | 0.615 | 0.332 |
| seymour | 0.044 | 0.035 | 176 | 563 | -0.243 | -0.195 | 0.611 |
| seymour1 | 0.003 | 0.003 | 329 | 885 | 0.146 | 0.145 | 0.682 |
| sp150x300d | 0.000 | 0.000 | 148 | 124 | 1.000 | 0.000 | 0.000 |
| supportcase18 | 0.081 | 0.081 | 178 | 1372 | 0.000 | -0.000 | 0.870 |
| supportcase26 | 0.224 | 0.231 | 11191 | 20287 | 0.031 | 0.031 | 0.465 |
| supportcase33 | 27.788 | 0.371 | 15 | 28 | -1.000 | -0.987 | -0.975 |
| supportcase40 | 0.086 | 0.094 | 50 | 111 | 0.087 | 0.087 | 0.589 |
| supportcase42 | 0.033 | 0.050 | 76 | 256 | 0.340 | 0.340 | 0.804 |
| swath1 | 0.000 | 0.000 | 311 | 372 | 1.000 | 0.000 | 0.000 |
| swath3 | 0.110 | 0.113 | 1442 | 2800 | 0.020 | 0.020 | 0.495 |
| timtab1 | 0.126 | 0.094 | 22112 | 25367 | -0.333 | -0.250 | -0.139 |
| tr12-30 | 0.002 | 0.002 | 8941 | 14896 | 0.019 | 0.019 | 0.394 |
| traininstance2 | ∞ | ∞ | 412 | 821 | 0.000 | 0.000 | 0.498 |
| traininstance6 | 29.355 | ∞ | 2549 | 6376 | 1.000 | 1.000 | 1.000 |
| trento1 | 3.885 | 3.885 | 4 | 7 | -0.000 | -0.000 | 0.429 |
| uct-subprob | 0.249 | 0.195 | 225 | 263 | -0.276 | -0.216 | -0.084 |
| var-smallemery-m6j6 | 0.062 | 0.062 | 95 | 224 | -0.002 | -0.002 | 0.575 |
| wachplan | 0.125 | 0.125 | 422 | 712 | 0.000 | 0.000 | 0.407 |
| Mean | — | — | 16538 | 19673 | **0.140** | -0.013 | **0.208** |

## I.4 MINLPLIB results

Table 8: Results on MINLPLIB (Bussieck et al., 2003) after 45s runtime. Note that we filter out problems in which less than 5 nodes were explored as those problems cannot gain meaningful advantages even with perfect node selection. "Name" refers to the instances name, "Gap Base/Ours" corresponds to the optimization gap achieved by the baseline and our method respectively (lower is better), "Nodes Base/Ours" to the number of explored Nodes by each method, and "Reward", "Utility" and "Utility Node" to the different performance measures as described in Section 5. For all three measures, higher is better.

| Name | Gap Ours | Gap Base | Nodes Ours | Nodes Base | Reward | Utility | Utility/Node |
|---|---|---|---|---|---|---|---|
| ball_mk4_05 | 0.000 | 0.000 | 1819 | 1869 | 1.000 | 0.000 | 0.000 |
| | | | | | | Continued on next page | |

| Name | Gap Ours | Gap Base | Nodes Ours | Nodes Base | Reward | Utility | Utility/Node |
|---|---|---|---|---|---|---|---|
| ball_mk4_10 | ∞ | ∞ | 31684 | 37656 | 0.000 | 0.000 | 0.159 |
| ball_mk4_15 | ∞ | ∞ | 1773 | 2415 | 0.000 | 0.000 | 0.266 |
| bayes2_20 | ∞ | ∞ | 3171 | 2719 | 0.000 | 0.000 | -0.143 |
| bayes2_30 | ∞ | ∞ | 4462 | 4992 | 0.000 | 0.000 | 0.106 |
| bayes2_50 | ∞ | ∞ | 2934 | 2530 | 0.000 | 0.000 | -0.138 |
| blend029 | 0.000 | 0.000 | 812 | 804 | 1.000 | 0.000 | 0.000 |
| blend146 | 0.097 | 0.105 | 12390 | 18066 | 0.075 | 0.075 | 0.365 |
| blend480 | 0.071 | 0.000 | 4878 | 6312 | -1.000 | -1.000 | -0.999 |
| blend531 | 0.000 | 0.000 | 3150 | 7161 | 1.000 | 0.000 | 0.000 |
| blend718 | 0.898 | 0.796 | 22652 | 26060 | -0.127 | -0.113 | 0.020 |
| blend721 | 0.000 | 0.000 | 4650 | 2708 | 1.000 | 0.000 | 0.000 |
| blend852 | 0.021 | 0.000 | 7726 | 5413 | -1.000 | -1.000 | -0.997 |
| camshape100 | 0.076 | 0.074 | 18839 | 22205 | -0.027 | -0.026 | 0.128 |
| camshape200 | 0.145 | 0.147 | 8199 | 9921 | 0.012 | 0.012 | 0.183 |
| camshape400 | 0.198 | 0.195 | 4324 | 5275 | -0.016 | -0.016 | 0.167 |
| camshape800 | 0.222 | 0.226 | 1504 | 1627 | 0.019 | 0.019 | 0.093 |
| cardqp_inlp | 1.436 | 1.660 | 4316 | 7232 | 0.135 | 0.135 | 0.484 |
| cardqp_iqp | 1.089 | 1.660 | 4766 | 7285 | 0.344 | 0.344 | 0.571 |
| carton7 | 0.000 | 0.000 | 55 | 73 | 1.000 | 0.000 | 0.000 |
| carton9 | 0.000 | 0.000 | 9848 | 7406 | 1.000 | 0.000 | 0.000 |
| catmix100 | ∞ | ∞ | 186 | 8750 | 0.000 | 0.000 | 0.979 |
| catmix200 | ∞ | ∞ | 123 | 3870 | 0.000 | 0.000 | 0.968 |
| catmix400 | ∞ | ∞ | 146 | 3498 | 0.000 | 0.000 | 0.958 |
| catmix800 | ∞ | ∞ | 75 | 333 | 0.000 | 0.000 | 0.775 |
| celar6-sub0 | ∞ | ∞ | 4 | 6 | 0.000 | 0.000 | 0.333 |
| chimera_k64ising-01 | 0.701 | 16.469 | 18 | 21 | 0.957 | 0.957 | 0.964 |
| chimera_k64maxcut-01 | 0.523 | 0.199 | 57 | 198 | -1.000 | -0.618 | 0.246 |
| chimera_k64maxcut-02 | 0.368 | 0.239 | 72 | 381 | -0.536 | -0.349 | 0.710 |
| chimera_lga-02 | 0.893 | 0.893 | 5 | 6 | 0.000 | 0.000 | 0.167 |
| chimera_mgw-c8-439-onc8-001 | 0.045 | 0.021 | 127 | 521 | -1.000 | -0.529 | 0.482 |
| chimera_mgw-c8-439-onc8-002 | 0.067 | 0.046 | 72 | 526 | -0.449 | -0.310 | 0.802 |
| chimera_mgw-c8-507-onc8-01 | 0.232 | 0.233 | 26 | 99 | 0.003 | 0.003 | 0.738 |
| chimera_mgw-c8-507-onc8-02 | 0.188 | 0.346 | 14 | 25 | 0.455 | 0.455 | 0.695 |
| chimera_mis-01 | 0.000 | 0.000 | 7 | 7 | 1.000 | 0.000 | 0.000 |
| chimera_mis-02 | 0.000 | 0.000 | 7 | 7 | 1.000 | 0.000 | 0.000 |
| chimera_rfr-01 | 1.029 | 1.153 | 70 | 61 | 0.108 | 0.108 | -0.023 |
| chimera_rfr-02 | 1.148 | 1.061 | 74 | 63 | -0.082 | -0.076 | -0.213 |
| chimera_selby-c8-onc8-01 | 0.436 | 0.224 | 34 | 111 | -0.941 | -0.485 | 0.406 |
| chimera_selby-c8-onc8-02 | 0.439 | 0.232 | 40 | 92 | -0.895 | -0.472 | 0.176 |
| clay0203m | 0.000 | 0.000 | 19 | 30 | 1.000 | 0.000 | 0.000 |
| clay0204m | 0.000 | 0.000 | 266 | 400 | 1.000 | 0.000 | 0.000 |
| clay0205m | 0.000 | 0.000 | 4058 | 3908 | 1.000 | 0.000 | 0.000 |
| clay0303m | 0.000 | 0.000 | 107 | 45 | 1.000 | 0.000 | 0.000 |
| clay0304m | 0.000 | 0.000 | 337 | 897 | 1.000 | 0.000 | 0.000 |
| clay0305m | 0.000 | 0.000 | 4057 | 4204 | 1.000 | 0.000 | 0.000 |
| color_lab3_3x0 | 1.445 | 1.725 | 320 | 576 | 0.162 | 0.162 | 0.534 |
| color_lab3_4x0 | 5.581 | 5.455 | 265 | 434 | -0.023 | -0.023 | 0.375 |
| crossdock_15x7 | 4.457 | 8.216 | 654 | 1080 | 0.458 | 0.458 | 0.672 |
| crossdock_15x8 | 8.578 | 84.148 | 391 | 717 | 0.898 | 0.898 | 0.944 |
| crudeoil_lee1_06 | 0.000 | 0.000 | 48 | 57 | 1.000 | 0.000 | 0.000 |
| crudeoil_lee1_07 | 0.000 | 0.000 | 57 | 92 | 1.000 | 0.000 | 0.000 |
| crudeoil_lee1_08 | 0.000 | 0.000 | 161 | 121 | 1.000 | 0.000 | 0.000 |
| crudeoil_lee1_09 | 0.000 | 0.000 | 107 | 99 | 1.000 | 0.000 | 0.000 |
| crudeoil_lee1_10 | 0.000 | 0.000 | 78 | 109 | 1.000 | 0.000 | 0.000 |
| crudeoil_lee2_05 | 0.000 | 0.000 | 10 | 11 | 1.000 | 0.000 | 0.000 |
| crudeoil_lee2_06 | 0.000 | 0.000 | 45 | 109 | 1.000 | 0.000 | 0.000 |
| crudeoil_lee2_07 | 0.000 | 0.000 | 286 | 81 | 1.000 | 0.000 | 0.000 |
| crudeoil_lee2_08 | 0.000 | 0.000 | 150 | 308 | 1.000 | 0.000 | 0.000 |
| crudeoil_lee2_09 | 0.142 | 0.015 | 44 | 41 | -1.000 | -0.897 | -0.904 |
| crudeoil_lee3_05 | 0.000 | 0.000 | 1435 | 1820 | 1.000 | 0.000 | 0.000 |
| crudeoil_lee3_06 | 0.057 | 0.013 | 352 | 1349 | -1.000 | -0.764 | -0.095 |
| crudeoil_lee4_05 | 0.000 | 0.000 | 306 | 118 | 1.000 | 0.000 | 0.000 |
| crudeoil_lee4_06 | 0.000 | 0.000 | 129 | 60 | 1.000 | 0.000 | 0.000 |
| crudeoil_lee4_07 | 0.000 | 0.000 | 193 | 89 | 1.000 | 0.000 | 0.000 |
| crudeoil_lee4_08 | 0.000 | 0.001 | 41 | 53 | 0.187 | 0.184 | 0.371 |
| crudeoil_li01 | 0.049 | 0.017 | 16819 | 11797 | -1.000 | -0.657 | -0.758 |
| crudeoil_li02 | 0.013 | 0.013 | 12172 | 10426 | -0.027 | -0.027 | -0.165 |
| crudeoil_li03 | ∞ | ∞ | 198 | 899 | 0.000 | 0.000 | 0.780 |
| crudeoil_li05 | 0.157 | 0.142 | 553 | 1031 | -0.104 | -0.095 | 0.408 |
| crudeoil_li06 | ∞ | ∞ | 41 | 322 | 0.000 | 0.000 | 0.873 |
| crudeoil_li11 | ∞ | ∞ | 20 | 70 | 0.000 | 0.000 | 0.714 |
| crudeoil_pooling_ct1 | 0.943 | 0.988 | 2415 | 6356 | 0.046 | 0.046 | 0.638 |
| crudeoil_pooling_ct2 | 0.000 | 0.000 | 1480 | 1589 | 1.000 | 0.000 | 0.000 |
| crudeoil_pooling_ct3 | 42.222 | 120.618 | 101 | 101 | 0.650 | 0.650 | 0.650 |
| crudeoil_pooling_ct4 | 0.000 | 0.000 | 7631 | 9217 | 0.365 | 0.153 | 0.041 |
| du-opt | 0.000 | 0.000 | 11282 | 14174 | 1.000 | 0.000 | 0.000 |
| du-opt5 | 0.000 | 0.000 | 83 | 60 | 1.000 | 0.000 | 0.000 |

| Name | Gap Ours | Gap Base | Nodes Ours | Nodes Base | Reward | Utility | Utility/Node |
|---|---|---|---|---|---|---|---|
| edgecross10-030 | 0.000 | 0.000 | 7 | 7 | 1.000 | 0.000 | 0.000 |
| edgecross10-040 | 0.000 | 0.000 | 30 | 39 | 1.000 | 0.000 | 0.000 |
| edgecross10-050 | 0.000 | 0.000 | 487 | 469 | 1.000 | 0.000 | 0.000 |
| edgecross10-060 | 0.000 | 0.000 | 2058 | 2138 | 1.000 | 0.000 | 0.000 |
| edgecross10-070 | 0.321 | 0.220 | 255 | 329 | -0.457 | -0.314 | -0.115 |
| edgecross10-080 | 0.077 | 0.077 | 352 | 668 | 0.001 | 0.001 | 0.474 |
| edgecross10-090 | 0.000 | 0.000 | 7 | 6 | 1.000 | 0.000 | 0.000 |
| edgecross14-039 | 0.000 | 0.000 | 624 | 731 | 1.000 | 0.000 | 0.000 |
| edgecross14-058 | 1.251 | 0.549 | 84 | 157 | -1.000 | -0.561 | -0.180 |
| edgecross14-078 | 1.843 | 1.865 | 12 | 14 | 0.012 | 0.012 | 0.153 |
| edgecross14-098 | 1.120 | 1.129 | 24 | 31 | 0.007 | 0.007 | 0.232 |
| edgecross14-117 | 0.963 | 0.947 | 9 | 17 | -0.017 | -0.017 | 0.462 |
| edgecross14-137 | 0.537 | 0.552 | 20 | 30 | 0.028 | 0.028 | 0.352 |
| edgecross14-156 | 0.338 | 0.353 | 13 | 13 | 0.042 | 0.042 | 0.042 |
| edgecross14-176 | 0.089 | 0.080 | 37 | 135 | -0.117 | -0.105 | 0.694 |
| edgecross20-040 | 0.000 | 0.000 | 71 | 57 | 1.000 | 0.000 | 0.000 |
| edgecross20-080 | 3.943 | 3.943 | 7 | 7 | 0.000 | 0.000 | 0.000 |
| edgecross22-048 | 0.615 | 0.000 | 56 | 81 | -1.000 | -1.000 | -1.000 |
| edgecross24-057 | 5.219 | 5.219 | 7 | 6 | 0.000 | 0.000 | -0.143 |
| elf | 0.000 | 0.000 | 115 | 112 | 1.000 | 0.000 | 0.000 |
| ex2_1_1 | 0.000 | 0.000 | 17 | 17 | 1.000 | 0.000 | 0.000 |
| ex2_1_10 | 0.000 | 0.000 | 13 | 11 | 1.000 | 0.000 | 0.000 |
| ex2_1_5 | 0.000 | 0.000 | 17 | 19 | 1.000 | 0.000 | 0.000 |
| ex2_1_6 | 0.000 | 0.000 | 13 | 13 | 1.000 | 0.000 | 0.000 |
| ex2_1_7 | 0.000 | 0.000 | 1523 | 1831 | 1.000 | 0.000 | 0.000 |
| ex2_1_8 | 0.000 | 0.000 | 75 | 93 | 1.000 | 0.000 | 0.000 |
| ex2_1_9 | 0.000 | 0.000 | 3735 | 3947 | 1.000 | 0.000 | 0.000 |
| ex3_1_1 | 0.000 | 0.000 | 405 | 271 | 1.000 | 0.000 | 0.000 |
| ex3_1_3 | 0.000 | 0.000 | 21 | 27 | 1.000 | 0.000 | 0.000 |
| ex3_1_4 | 0.000 | 0.000 | 23 | 23 | 1.000 | 0.000 | 0.000 |
| ex4 | 0.000 | 0.000 | 23 | 29 | 1.000 | 0.000 | 0.000 |
| ex5_2_2_case1 | 0.000 | 0.000 | 39 | 19 | 1.000 | 0.000 | 0.000 |
| ex5_2_2_case2 | 0.000 | 0.000 | 57 | 31 | 1.000 | 0.000 | 0.000 |
| ex5_2_4 | 0.000 | 0.000 | 251 | 227 | 1.000 | 0.000 | 0.000 |
| ex5_2_5 | 0.359 | 0.346 | 30403 | 33492 | -0.038 | -0.036 | 0.058 |
| ex5_3_2 | 0.000 | 0.000 | 33 | 31 | 1.000 | 0.000 | 0.000 |
| ex5_3_3 | 0.339 | 0.331 | 29464 | 31558 | -0.024 | -0.024 | 0.044 |
| ex5_4_2 | 0.000 | 0.000 | 41 | 35 | 1.000 | 0.000 | 0.000 |
| ex8_3_2 | 23.252 | 23.608 | 8907 | 8680 | 0.015 | 0.015 | -0.011 |
| ex8_3_3 | 23.004 | 23.004 | 9636 | 10365 | 0.000 | 0.000 | 0.070 |
| ex8_3_4 | 1.817 | 1.793 | 9447 | 9563 | -0.013 | -0.013 | -0.001 |
| ex8_3_5 | 143.677 | 143.677 | 9427 | 9699 | 0.000 | 0.000 | 0.028 |
| ex8_3_8 | 2.071 | 2.071 | 2293 | 3677 | 0.000 | 0.000 | 0.376 |
| ex8_3_9 | 12.106 | 12.106 | 14272 | 17310 | 0.000 | -0.000 | 0.176 |
| ex8_4_1 | 0.000 | 0.000 | 670 | 650 | 1.000 | 0.000 | 0.000 |
| ex9_2_3 | 0.000 | 0.000 | 25 | 31 | 1.000 | 0.000 | 0.000 |
| ex9_2_5 | 0.000 | 0.000 | 27 | 29 | 1.000 | 0.000 | 0.000 |
| ex9_2_7 | 0.000 | 0.000 | 11 | 11 | 1.000 | 0.000 | 0.000 |
| faclay20h | 1.727 | 1.727 | 16 | 15 | 0.000 | 0.000 | -0.062 |
| faclay25 | 2.468 | 2.468 | 6 | 6 | 0.000 | 0.000 | 0.000 |
| forest | 0.003 | 0.020 | 29002 | 25913 | 0.860 | 0.859 | 0.831 |
| gabriel01 | 0.139 | 0.139 | 6753 | 9744 | -0.000 | -0.000 | 0.307 |
| gabriel02 | 0.556 | 0.585 | 1107 | 1675 | 0.050 | 0.050 | 0.372 |
| gabriel04 | ∞ | 1.308 | 129 | 285 | -1.000 | -1.000 | -1.000 |
| gabriel05 | ∞ | ∞ | 141 | 326 | 0.000 | 0.000 | 0.567 |
| gasprod_sarawak01 | 0.000 | 0.000 | 11 | 6 | 1.000 | 0.000 | 0.000 |
| gasprod_sarawak16 | 0.004 | 0.009 | 506 | 1052 | 0.585 | 0.585 | 0.800 |
| genpooling_lee1 | 0.000 | 0.000 | 690 | 676 | 1.000 | 0.000 | 0.000 |
| genpooling_lee2 | 0.000 | 0.000 | 1299 | 2989 | 1.000 | 0.000 | 0.000 |
| genpooling_meyer04 | 0.957 | 0.691 | 12855 | 17889 | -0.385 | -0.278 | 0.005 |
| genpooling_meyer10 | 1.276 | 1.385 | 1910 | 2815 | 0.078 | 0.078 | 0.375 |
| genpooling_meyer15 | 6.080 | 0.691 | 97 | 413 | -1.000 | -0.886 | -0.516 |
| graphpart_2g-0099-9211 | 0.000 | 0.000 | 18 | 14 | 1.000 | 0.000 | 0.000 |
| graphpart_2pm-0077-0777 | 0.000 | 0.000 | 5 | 6 | 1.000 | 0.000 | 0.000 |
| graphpart_2pm-0088-0888 | 0.000 | 0.000 | 9 | 7 | 1.000 | 0.000 | 0.000 |
| graphpart_2pm-0099-0999 | 0.000 | 0.000 | 16 | 12 | 1.000 | 0.000 | 0.000 |
| graphpart_3g-0334-0334 | 0.000 | 0.000 | 21 | 41 | 1.000 | 0.000 | 0.000 |
| graphpart_3g-0344-0344 | 0.000 | 0.000 | 61 | 19 | 1.000 | 0.000 | 0.000 |
| graphpart_3g-0444-0444 | 0.000 | 0.000 | 424 | 562 | 1.000 | 0.000 | 0.000 |
| graphpart_3pm-0244-0244 | 0.000 | 0.000 | 21 | 15 | 1.000 | 0.000 | 0.000 |
| graphpart_3pm-0334-0334 | 0.000 | 0.000 | 20 | 38 | 1.000 | 0.000 | 0.000 |
| graphpart_3pm-0344-0344 | 0.000 | 0.000 | 590 | 619 | 1.000 | 0.000 | 0.000 |
| graphpart_3pm-0444-0444 | 0.058 | 0.000 | 755 | 1348 | -1.000 | -1.000 | -1.000 |
| graphpart_clique-20 | 0.000 | 0.000 | 22 | 24 | 1.000 | 0.000 | 0.000 |
| graphpart_clique-30 | 0.000 | 0.000 | 421 | 337 | 1.000 | 0.000 | 0.000 |
| graphpart_clique-40 | 1.018 | 0.920 | 297 | 609 | -0.106 | -0.096 | 0.461 |
| graphpart_clique-50 | 5.638 | 6.032 | 97 | 191 | 0.065 | 0.065 | 0.525 |

| Name | Gap Ours | Gap Base | Nodes Ours | Nodes Base | Reward | Utility | Utility/Node |
|---|---|---|---|---|---|---|---|
| graphpart_clique-60 | 17.434 | 9.335 | 109 | 204 | -0.868 | -0.465 | 0.002 |
| graphpart_clique-70 | 30.409 | 35.053 | 16 | 27 | 0.132 | 0.132 | 0.486 |
| haverly | 0.000 | 0.000 | 45 | 57 | 1.000 | 0.000 | 0.000 |
| himmel16 | 0.000 | 0.000 | 2193 | 2089 | 1.000 | 0.000 | 0.000 |
| house | 0.000 | 0.000 | 58675 | 58399 | 1.000 | 0.000 | 0.000 |
| hvb11 | 0.018 | 0.182 | 19172 | 15631 | 0.899 | 0.899 | 0.875 |
| hydroenergy1 | 0.007 | 0.007 | 15060 | 18149 | -0.088 | -0.081 | 0.095 |
| hydroenergy2 | 0.016 | 0.016 | 4834 | 6712 | 0.038 | 0.038 | 0.306 |
| hydroenergy3 | 0.022 | 0.023 | 565 | 1060 | 0.006 | 0.006 | 0.470 |
| ising2_5-300_5555 | 0.508 | 0.407 | 57 | 220 | -0.248 | -0.199 | 0.677 |
| kall_circles_c6a | 3.180 | 2.094 | 42813 | 46497 | -0.519 | -0.342 | -0.285 |
| kall_circles_c6b | 2.635 | 1.452 | 38722 | 45596 | -0.815 | -0.449 | -0.351 |
| kall_circles_c6c | $\infty$ | $\infty$ | 33357 | 36374 | 0.000 | 0.000 | 0.083 |
| kall_circles_c7a | 1.482 | 1.376 | 38682 | 43723 | -0.077 | -0.072 | 0.047 |
| kall_circles_c8a | $\infty$ | $\infty$ | 32114 | 36262 | 0.000 | 0.000 | 0.114 |
| kall_circlespolygons_c1p12 | 0.000 | 0.000 | 44439 | 64102 | -1.000 | -0.733 | -0.106 |
| kall_circlespolygons_c1p13 | 0.000 | 0.000 | 8621 | 7914 | 1.000 | 0.000 | 0.000 |
| kall_circlespolygons_c1p5a | $\infty$ | $\infty$ | 12369 | 13200 | 0.000 | 0.000 | 0.063 |
| kall_circlespolygons_c1p6a | $\infty$ | $\infty$ | 404 | 628 | 0.000 | 0.000 | 0.357 |
| kall_circlesrectangles_c1r12 | 0.000 | 0.000 | 42587 | 48285 | 0.121 | 0.114 | 0.061 |
| kall_circlesrectangles_c1r13 | 0.000 | 0.000 | 4372 | 3739 | 1.000 | 0.000 | 0.000 |
| kall_circlesrectangles_c6r1 | $\infty$ | $\infty$ | 5850 | 7908 | 0.000 | 0.000 | 0.260 |
| kall_circlesrectangles_c6r29 | $\infty$ | $\infty$ | 4181 | 5220 | 0.000 | 0.000 | 0.199 |
| kall_circlesrectangles_c6r39 | $\infty$ | $\infty$ | 2570 | 2966 | 0.000 | 0.000 | 0.134 |
| kall_congruentcircles_c31 | 0.000 | 0.000 | 101 | 95 | 1.000 | 0.000 | 0.000 |
| kall_congruentcircles_c32 | 0.000 | 0.000 | 133 | 139 | 1.000 | 0.000 | 0.000 |
| kall_congruentcircles_c41 | 0.000 | 0.000 | 27 | 31 | 1.000 | 0.000 | 0.000 |
| kall_congruentcircles_c42 | 0.000 | 0.000 | 205 | 125 | 1.000 | 0.000 | 0.000 |
| kall_congruentcircles_c51 | 0.000 | 0.000 | 4197 | 4987 | 1.000 | 0.000 | 0.000 |
| kall_congruentcircles_c52 | 0.000 | 0.000 | 1767 | 1446 | 1.000 | 0.000 | 0.000 |
| kall_congruentcircles_c61 | 0.000 | 0.000 | 27338 | 35199 | 1.000 | 0.000 | 0.000 |
| kall_congruentcircles_c62 | 0.000 | 0.000 | 2879 | 6037 | 1.000 | 0.000 | 0.000 |
| kall_congruentcircles_c63 | 0.000 | 0.000 | 2043 | 1729 | 1.000 | 0.000 | 0.000 |
| kall_congruentcircles_c71 | $\infty$ | $\infty$ | 39102 | 43349 | 0.000 | 0.000 | 0.098 |
| kall_congruentcircles_c72 | 0.000 | 0.000 | 14686 | 14089 | 1.000 | 0.000 | 0.000 |
| kall_diffcircles_10 | 2.276 | 4.054 | 32475 | 41241 | 0.439 | 0.439 | 0.558 |
| kall_diffcircles_5a | 0.000 | 0.000 | 2020 | 1218 | 1.000 | 0.000 | 0.000 |
| kall_diffcircles_5b | 0.000 | 0.000 | 6360 | 5774 | 1.000 | 0.000 | 0.000 |
| kall_diffcircles_6 | 0.000 | 0.000 | 2827 | 2383 | 1.000 | 0.000 | 0.000 |
| kall_diffcircles_7 | 0.000 | 0.000 | 9408 | 9518 | 1.000 | 0.000 | 0.000 |
| kall_diffcircles_8 | 0.406 | 0.219 | 48924 | 57747 | -0.851 | -0.460 | -0.362 |
| kall_diffcircles_9 | 1.676 | 1.052 | 42056 | 48915 | -0.594 | -0.373 | -0.270 |
| knp3-12 | 1.846 | 1.963 | 1987 | 2132 | 0.060 | 0.060 | 0.124 |
| lop97ic | $\infty$ | $\infty$ | 19 | 33 | 0.000 | 0.000 | 0.424 |
| lop97icx | 0.008 | 0.000 | 3041 | 1711 | -1.000 | -0.999 | -0.998 |
| maxcsp-langford-3-11 | $\infty$ | $\infty$ | 1356 | 4038 | 0.000 | 0.000 | 0.664 |
| ndcc12 | $\infty$ | $\infty$ | 1394 | 3975 | 0.000 | 0.000 | 0.649 |
| ndcc12persp | $\infty$ | $\infty$ | 1092 | 2994 | 0.000 | 0.000 | 0.635 |
| ndcc13 | $\infty$ | $\infty$ | 298 | 787 | 0.000 | 0.000 | 0.621 |
| ndcc13persp | 0.536 | 0.546 | 2982 | 5662 | 0.018 | 0.018 | 0.483 |
| ndcc14 | 1.030 | 1.048 | 234 | 499 | 0.018 | 0.018 | 0.539 |
| ndcc14persp | 1.044 | 1.080 | 572 | 1052 | 0.033 | 0.033 | 0.474 |
| ndcc15 | $\infty$ | $\infty$ | 1293 | 2120 | 0.000 | 0.000 | 0.390 |
| ndcc15persp | $\infty$ | $\infty$ | 5227 | 6549 | 0.000 | 0.000 | 0.202 |
| ndcc16 | $\infty$ | $\infty$ | 407 | 396 | 0.000 | 0.000 | -0.027 |
| ndcc16persp | $\infty$ | $\infty$ | 1035 | 2183 | 0.000 | 0.000 | 0.526 |
| netmod_dol2 | 0.047 | 0.000 | 112 | 250 | -1.000 | -1.000 | -1.000 |
| netmod_kar1 | 0.000 | 0.000 | 425 | 285 | 1.000 | 0.000 | 0.000 |
| netmod_kar2 | 0.000 | 0.000 | 275 | 285 | 1.000 | 0.000 | 0.000 |
| nous1 | 0.000 | 0.000 | 3092 | 2816 | 1.000 | 0.000 | 0.000 |
| nous2 | 0.000 | 0.000 | 81 | 71 | 1.000 | 0.000 | 0.000 |
| nuclearvb | $\infty$ | $\infty$ | 1821 | 3817 | 0.000 | 0.000 | 0.523 |
| nuclearvc | $\infty$ | $\infty$ | 1905 | 1530 | 0.000 | 0.000 | -0.197 |
| nuclearvd | $\infty$ | $\infty$ | 3781 | 2521 | 0.000 | 0.000 | -0.333 |
| nuclearve | $\infty$ | $\infty$ | 877 | 5464 | 0.000 | 0.000 | 0.839 |
| nuclearvf | $\infty$ | $\infty$ | 256 | 3596 | 0.000 | 0.000 | 0.929 |
| nvs13 | 0.000 | 0.000 | 9 | 9 | 1.000 | 0.000 | 0.000 |
| nvs17 | 0.000 | 0.000 | 89 | 78 | 1.000 | 0.000 | 0.000 |
| nvs18 | 0.000 | 0.000 | 121 | 75 | 1.000 | 0.000 | 0.000 |
| nvs19 | 0.000 | 0.000 | 161 | 154 | 1.000 | 0.000 | 0.000 |
| nvs23 | 0.000 | 0.000 | 465 | 523 | 1.000 | 0.000 | 0.000 |
| nvs24 | 0.000 | 0.000 | 2060 | 1944 | 1.000 | 0.000 | 0.000 |
| p_ball_10b_5p_2d_m | 0.000 | 0.000 | 353 | 326 | 1.000 | 0.000 | 0.000 |
| p_ball_10b_5p_3d_m | 0.000 | 0.000 | 1204 | 1032 | 1.000 | 0.000 | 0.000 |
| p_ball_10b_5p_4d_m | 0.000 | 0.000 | 1424 | 1765 | 1.000 | 0.000 | 0.000 |
| p_ball_10b_7p_3d_m | 0.000 | 0.000 | 6178 | 6151 | 1.000 | 0.000 | 0.000 |
| p_ball_15b_5p_2d_m | 0.000 | 0.000 | 1377 | 2068 | 1.000 | 0.000 | 0.000 |

| Name | Gap Ours | Gap Base | Nodes Ours | Nodes Base | Reward | Utility | Utility/Node |
|---|---|---|---|---|---|---|---|
| p_ball_20b_5p_2d_m | 0.000 | 0.000 | 1610 | 2039 | 1.000 | 0.000 | 0.000 |
| p_ball_20b_5p_3d_m | 0.000 | 0.000 | 10647 | 11510 | 1.000 | 0.000 | 0.000 |
| p_ball_30b_10p_2d_m | ∞ | ∞ | 3795 | 4965 | 0.000 | 0.000 | 0.236 |
| p_ball_30b_5p_2d_m | 0.000 | 0.000 | 2827 | 3275 | 1.000 | 0.000 | 0.000 |
| p_ball_30b_5p_3d_m | 0.000 | 0.000 | 10150 | 11489 | 1.000 | 0.000 | 0.000 |
| p_ball_30b_7p_2d_m | ∞ | ∞ | 8511 | 11906 | 0.000 | 0.000 | 0.285 |
| p_ball_40b_5p_3d_m | ∞ | ∞ | 9620 | 13718 | 0.000 | 0.000 | 0.299 |
| p_ball_40b_5p_4d_m | ∞ | ∞ | 8100 | 11826 | 0.000 | 0.000 | 0.315 |
| pedigree_ex485 | 0.019 | 0.019 | 315 | 962 | 0.030 | 0.030 | 0.682 |
| pedigree_ex485_2 | 0.000 | 0.000 | 121 | 344 | 1.000 | 0.000 | 0.000 |
| pedigree_sim400 | 0.061 | 0.053 | 1094 | 1533 | -0.156 | -0.135 | 0.175 |
| pedigree_sp_top4_250 | 0.053 | 0.036 | 61 | 173 | -0.482 | -0.325 | 0.477 |
| pedigree_sp_top4_300 | 0.014 | 0.015 | 294 | 670 | 0.014 | 0.014 | 0.567 |
| pedigree_sp_top4_350tr | 0.000 | 0.014 | 365 | 1096 | 1.000 | 0.999 | 1.000 |
| pedigree_sp_top5_250 | 0.050 | 0.057 | 28 | 39 | 0.125 | 0.125 | 0.372 |
| pinene200 | ∞ | ∞ | 12 | 12 | 0.000 | 0.000 | 0.000 |
| pointpack06 | 0.000 | 0.000 | 2099 | 2051 | 1.000 | 0.000 | 0.000 |
| pointpack08 | 0.015 | 0.000 | 35620 | 34315 | -1.000 | -0.999 | -0.978 |
| pointpack10 | 0.612 | 0.613 | 18366 | 22179 | 0.001 | 0.001 | 0.173 |
| pointpack12 | 0.854 | 0.839 | 15197 | 17796 | -0.018 | -0.018 | 0.131 |
| pointpack14 | 1.535 | 1.537 | 8919 | 9550 | 0.001 | 0.001 | 0.067 |
| pooling_adhya1pq | 0.000 | 0.000 | 383 | 365 | 1.000 | 0.000 | 0.000 |
| pooling_adhya1stp | 0.000 | 0.000 | 737 | 638 | 1.000 | 0.000 | 0.000 |
| pooling_adhya1tp | 0.000 | 0.000 | 611 | 806 | 1.000 | 0.000 | 0.000 |
| pooling_adhya2pq | 0.000 | 0.000 | 569 | 588 | 1.000 | 0.000 | 0.000 |
| pooling_adhya2stp | 0.000 | 0.000 | 832 | 934 | 1.000 | 0.000 | 0.000 |
| pooling_adhya2tp | 0.000 | 0.000 | 345 | 288 | 1.000 | 0.000 | 0.000 |
| pooling_adhya3pq | 0.000 | 0.000 | 377 | 289 | 1.000 | 0.000 | 0.000 |
| pooling_adhya3stp | 0.000 | 0.000 | 834 | 1078 | 1.000 | 0.000 | 0.000 |
| pooling_adhya3tp | 0.000 | 0.000 | 675 | 585 | 1.000 | 0.000 | 0.000 |
| pooling_adhya4pq | 0.000 | 0.000 | 274 | 150 | 1.000 | 0.000 | 0.000 |
| pooling_adhya4stp | 0.000 | 0.000 | 385 | 686 | 1.000 | 0.000 | 0.000 |
| pooling_adhya4tp | 0.000 | 0.000 | 317 | 387 | 1.000 | 0.000 | 0.000 |
| pooling_bental5stp | 0.000 | 0.000 | 2818 | 4434 | 1.000 | 0.000 | 0.000 |
| pooling_digabel16 | 0.000 | 0.000 | 27577 | 35207 | -1.000 | -0.715 | -0.160 |
| pooling_digabel18 | 0.013 | 0.008 | 4109 | 5110 | -0.496 | -0.331 | -0.168 |
| pooling_digabel19 | 0.001 | 0.001 | 14953 | 18095 | 0.168 | 0.166 | 0.267 |
| pooling_foulds2stp | 0.000 | 0.000 | 36 | 25 | 1.000 | 0.000 | 0.000 |
| pooling_foulds3stp | 0.000 | 0.000 | 1084 | 416 | 1.000 | 0.000 | 0.000 |
| pooling_foulds4stp | 0.000 | 0.000 | 717 | 339 | 1.000 | 0.000 | 0.000 |
| pooling_foulds5stp | 0.019 | 0.000 | 1808 | 2741 | -1.000 | -0.999 | -0.999 |
| pooling_haverly2stp | 0.000 | 0.000 | 10 | 12 | 1.000 | 0.000 | 0.000 |
| pooling_rt2pq | 0.000 | 0.000 | 237 | 431 | 1.000 | 0.000 | 0.000 |
| pooling_rt2stp | 0.000 | 0.000 | 109 | 195 | 1.000 | 0.000 | 0.000 |
| pooling_rt2tp | 0.000 | 0.000 | 53 | 57 | 1.000 | 0.000 | 0.000 |
| pooling_sppa0pq | 0.038 | 0.031 | 2424 | 3666 | -0.230 | -0.187 | 0.187 |
| pooling_sppa0stp | 2.829 | 2.865 | 2577 | 3068 | 0.012 | 0.012 | 0.170 |
| pooling_sppa0tp | 0.179 | 0.183 | 2804 | 3623 | 0.021 | 0.021 | 0.242 |
| pooling_sppa5pq | 0.037 | 0.018 | 709 | 781 | -0.995 | -0.499 | -0.448 |
| pooling_sppa5stp | 3.959 | 3.959 | 220 | 278 | 0.000 | 0.000 | 0.209 |
| pooling_sppa5tp | 1.579 | 1.579 | 299 | 448 | 0.000 | 0.000 | 0.333 |
| pooling_sppa9pq | 0.007 | 0.007 | 222 | 295 | 0.000 | 0.000 | 0.247 |
| pooling_sppb0pq | 0.098 | 0.098 | 223 | 301 | 0.000 | -0.000 | 0.259 |
| popdynm100 | ∞ | ∞ | 7556 | 11105 | 0.000 | 0.000 | 0.320 |
| popdynm25 | ∞ | ∞ | 14627 | 19046 | 0.000 | 0.000 | 0.232 |
| popdynm50 | ∞ | ∞ | 12085 | 15252 | 0.000 | 0.000 | 0.208 |
| portfol_classical050_1 | 0.000 | 0.000 | 651 | 817 | 1.000 | 0.000 | 0.000 |
| portfol_classical200_2 | 0.141 | 0.125 | 396 | 491 | -0.134 | -0.118 | 0.086 |
| portfol_robust050_34 | 0.000 | 0.000 | 94 | 49 | 1.000 | 0.000 | 0.000 |
| portfol_robust100_09 | 0.000 | 0.000 | 489 | 361 | 1.000 | 0.000 | 0.000 |
| portfol_robust200_03 | 0.182 | 0.189 | 95 | 75 | 0.034 | 0.034 | -0.183 |
| portfol_shortfall050_68 | 0.000 | 0.000 | 467 | 375 | 1.000 | 0.000 | 0.000 |
| portfol_shortfall100_04 | 0.010 | 0.010 | 595 | 1398 | -0.055 | -0.052 | 0.551 |
| portfol_shortfall200_05 | 0.033 | 0.028 | 224 | 232 | -0.169 | -0.145 | -0.114 |
| powerflow0009r | 0.000 | 0.000 | 15230 | 13141 | -1.000 | -0.037 | -0.003 |
| powerflow0014r | 0.001 | 0.001 | 8052 | 8041 | 0.368 | 0.366 | 0.346 |
| powerflow0030r | 0.023 | 0.034 | 369 | 403 | 0.328 | 0.328 | 0.384 |
| powerflow0039r | 0.017 | 0.016 | 212 | 224 | -0.058 | -0.054 | -0.001 |
| product | 0.028 | 0.034 | 236 | 650 | 0.197 | 0.197 | 0.708 |
| qap | 198.418 | ∞ | 709 | 3352 | 1.000 | 1.000 | 1.000 |
| qapw | 351.271 | ∞ | 874 | 2437 | 1.000 | 1.000 | 1.000 |
| qp3 | ∞ | ∞ | 29875 | 32155 | 0.000 | 0.000 | 0.071 |
| qspp_0_10_0_1_10_1 | 0.849 | 1.238 | 3860 | 3982 | 0.314 | 0.314 | 0.335 |
| qspp_0_11_0_1_10_1 | 1.071 | 1.886 | 1314 | 3036 | 0.432 | 0.432 | 0.754 |
| qspp_0_12_0_1_10_1 | 1.674 | 2.102 | 794 | 1847 | 0.204 | 0.203 | 0.658 |
| qspp_0_13_0_1_10_1 | 1.893 | 4.660 | 935 | 1380 | 0.594 | 0.594 | 0.725 |
| qspp_0_14_0_1_10_1 | 3.038 | 3.200 | 299 | 1081 | 0.050 | 0.050 | 0.737 |

| Name | Gap Ours | Gap Base | Nodes Ours | Nodes Base | Reward | Utility | Utility/Node |
|---|---|---|---|---|---|---|---|
| qspp_0_15_0_1_10_1 | 4.356 | 4.293 | 229 | 544 | -0.015 | -0.015 | 0.573 |
| ringpack_10_1 | 0.082 | 1.000 | 5346 | 6348 | 0.918 | 0.918 | 0.931 |
| ringpack_10_2 | 0.082 | 0.811 | 5402 | 6366 | 0.899 | 0.899 | 0.914 |
| ringpack_20_1 | 1.551 | 3.527 | 525 | 492 | 0.560 | 0.560 | 0.531 |
| ringpack_20_2 | 9.000 | 9.000 | 239 | 183 | 0.000 | 0.000 | -0.234 |
| ringpack_20_3 | 6.251 | 6.251 | 272 | 243 | 0.000 | 0.000 | -0.107 |
| ringpack_30_2 | 14.000 | 14.000 | 36 | 49 | 0.000 | 0.000 | 0.265 |
| sep1 | 0.000 | 0.000 | 39 | 29 | 1.000 | 0.000 | 0.000 |
| slay04h | 0.000 | 0.000 | 8 | 8 | 1.000 | 0.000 | 0.000 |
| slay04m | 0.000 | 0.000 | 7 | 7 | 1.000 | 0.000 | 0.000 |
| slay05h | 0.000 | 0.000 | 64 | 119 | 1.000 | 0.000 | 0.000 |
| slay06h | 0.000 | 0.000 | 120 | 208 | 1.000 | 0.000 | 0.000 |
| slay06m | 0.000 | 0.000 | 8 | 8 | 1.000 | 0.000 | 0.000 |
| slay07h | 0.000 | 0.000 | 420 | 952 | 1.000 | 0.000 | 0.000 |
| slay07m | 0.000 | 0.000 | 218 | 501 | 1.000 | 0.000 | 0.000 |
| slay08h | 0.000 | 0.000 | 513 | 1181 | 1.000 | 0.000 | 0.000 |
| slay08m | 0.000 | 0.000 | 193 | 554 | 1.000 | 0.000 | 0.000 |
| slay09h | 0.104 | 0.135 | 612 | 488 | 0.229 | 0.229 | 0.033 |
| slay09m | 0.000 | 0.000 | 324 | 212 | 1.000 | 0.000 | 0.000 |
| slay10h | 0.103 | 0.407 | 703 | 451 | 0.746 | 0.745 | 0.603 |
| slay10m | 0.000 | 0.000 | 3933 | 4138 | 1.000 | 0.000 | 0.000 |
| smallinvDAXr1b010-011 | 0.000 | 0.000 | 324 | 264 | 1.000 | 0.000 | 0.000 |
| smallinvDAXr1b020-022 | 0.000 | 0.000 | 657 | 906 | 1.000 | 0.000 | 0.000 |
| smallinvDAXr1b050-055 | 0.000 | 0.000 | 6083 | 4430 | 1.000 | 0.000 | 0.000 |
| smallinvDAXr1b100-110 | 0.000 | 0.000 | 15366 | 34917 | 1.000 | 0.000 | 0.000 |
| smallinvDAXr1b150-165 | 0.000 | 0.001 | 26952 | 40900 | 1.000 | 0.986 | 0.730 |
| smallinvDAXr1b200-220 | 0.000 | 0.001 | 38238 | 46021 | 0.348 | 0.342 | 0.269 |
| smallinvDAXr2b010-011 | 0.000 | 0.000 | 254 | 358 | 1.000 | 0.000 | 0.000 |
| smallinvDAXr2b020-022 | 0.000 | 0.000 | 1204 | 2016 | 1.000 | 0.000 | 0.000 |
| smallinvDAXr2b050-055 | 0.000 | 0.000 | 7868 | 6682 | 1.000 | 0.000 | 0.000 |
| smallinvDAXr2b100-110 | 0.000 | 0.000 | 12971 | 14333 | 1.000 | 0.000 | 0.000 |
| smallinvDAXr2b150-165 | 0.000 | 0.000 | 39670 | 68543 | 1.000 | 0.966 | 0.421 |
| smallinvDAXr2b200-220 | 0.000 | 0.000 | 712 | 651 | 1.000 | 0.000 | 0.000 |
| smallinvDAXr3b010-011 | 0.000 | 0.000 | 260 | 358 | 1.000 | 0.000 | 0.000 |
| smallinvDAXr3b020-022 | 0.000 | 0.000 | 1676 | 906 | 1.000 | 0.000 | 0.000 |
| smallinvDAXr3b050-055 | 0.000 | 0.000 | 5716 | 5024 | 1.000 | 0.000 | 0.000 |
| smallinvDAXr3b100-110 | 0.000 | 0.000 | 39948 | 13726 | 1.000 | 0.000 | 0.000 |
| smallinvDAXr3b150-165 | 0.000 | 0.000 | 34109 | 22132 | 1.000 | 0.000 | 0.000 |
| smallinvDAXr3b200-220 | 0.000 | 0.000 | 1078 | 433 | 1.000 | 0.000 | 0.000 |
| smallinvDAXr4b010-011 | 0.000 | 0.000 | 272 | 292 | 1.000 | 0.000 | 0.000 |
| smallinvDAXr4b020-022 | 0.000 | 0.000 | 1078 | 990 | 1.000 | 0.000 | 0.000 |
| smallinvDAXr4b050-055 | 0.000 | 0.000 | 3098 | 2666 | 1.000 | 0.000 | 0.000 |
| smallinvDAXr4b100-110 | 0.000 | 0.000 | 17899 | 26316 | 1.000 | 0.000 | 0.000 |
| smallinvDAXr4b150-165 | 0.000 | 0.000 | 32042 | 56419 | 1.000 | 0.000 | 0.000 |
| smallinvDAXr4b200-220 | 0.000 | 0.000 | 935 | 612 | 1.000 | 0.000 | 0.000 |
| smallinvDAXr5b010-011 | 0.000 | 0.000 | 242 | 381 | 1.000 | 0.000 | 0.000 |
| smallinvDAXr5b020-022 | 0.000 | 0.000 | 1798 | 884 | 1.000 | 0.000 | 0.000 |
| smallinvDAXr5b050-055 | 0.000 | 0.000 | 4276 | 3312 | 1.000 | 0.000 | 0.000 |
| smallinvDAXr5b100-110 | 0.000 | 0.000 | 37028 | 72501 | 1.000 | 0.980 | 0.570 |
| smallinvDAXr5b150-165 | 0.000 | 0.000 | 40757 | 78031 | 1.000 | 0.966 | 0.414 |
| smallinvDAXr5b200-220 | 0.000 | 0.000 | 783 | 585 | 1.000 | 0.000 | 0.000 |
| sonet22v5 | 3.752 | 2.911 | 105 | 356 | -0.289 | -0.224 | 0.620 |
| sonet23v4 | 1.407 | 1.366 | 79 | 181 | -0.030 | -0.029 | 0.550 |
| sonet24v5 | 4.070 | 3.914 | 21 | 212 | -0.040 | -0.038 | 0.897 |
| sonet25v6 | 5.161 | 4.812 | 10 | 45 | -0.072 | -0.068 | 0.762 |
| sonetgr17 | 2.252 | 2.602 | 400 | 1247 | 0.134 | 0.134 | 0.722 |
| space25 | $\infty$ | $\infty$ | 154 | 143 | 0.000 | 0.000 | -0.071 |
| spectra2 | 0.000 | 0.000 | 8 | 8 | 1.000 | 0.000 | 0.000 |
| squfl010-025 | 0.000 | 0.000 | 71985 | 75945 | 0.692 | 0.000 | 0.000 |
| squfl010-040 | 0.000 | 0.000 | 18478 | 20101 | 0.529 | 0.000 | 0.000 |
| squfl010-080 | 0.000 | 0.000 | 4509 | 8339 | 0.568 | -0.000 | 0.000 |
| squfl010-080persp | 0.000 | 0.000 | 6 | 6 | 1.000 | 0.000 | 0.000 |
| squfl015-060 | 0.000 | 0.000 | 7372 | 10223 | 0.608 | -0.000 | 0.000 |
| squfl015-060persp | 0.000 | 0.000 | 6 | 6 | 1.000 | 0.000 | 0.000 |
| squfl015-080 | 0.000 | 0.001 | 3475 | 6667 | 1.000 | 0.993 | 0.976 |
| squfl020-040 | 0.000 | 0.000 | 8358 | 10679 | 0.570 | 0.000 | 0.000 |
| squfl020-050 | 0.000 | 0.000 | 4094 | 8025 | 0.365 | -0.000 | 0.000 |
| squfl020-150 | 0.014 | 0.014 | 9 | 7 | 0.000 | 0.000 | -0.222 |
| squfl020-150persp | 0.000 | 0.000 | 16 | 16 | 1.000 | 0.000 | 0.000 |
| squfl025-025 | 0.000 | 0.000 | 15093 | 11992 | 0.997 | -0.000 | -0.000 |
| squfl025-025persp | 0.000 | 0.000 | 12 | 12 | 1.000 | 0.000 | 0.000 |
| squfl025-030 | 0.000 | 0.000 | 5523 | 14798 | 1.000 | 0.000 | 0.000 |
| squfl025-030persp | 0.000 | 0.000 | 6 | 6 | 1.000 | 0.000 | 0.000 |
| squfl025-040 | 0.000 | 0.000 | 6438 | 7860 | 0.519 | 0.000 | 0.000 |
| squfl025-040persp | 0.000 | 0.000 | 12 | 12 | 1.000 | 0.000 | 0.000 |
| squfl030-100 | 0.000 | 0.000 | 1291 | 1402 | 0.289 | 0.000 | 0.000 |
| squfl040-080 | 0.000 | 0.001 | 1034 | 1477 | 1.000 | 0.983 | 0.983 |

Continued on next page

| Name | Gap Ours | Gap Base | Nodes Ours | Nodes Base | Reward | Utility | Utility/Node |
|---|---|---|---|---|---|---|---|
| squfl040-080persp | 0.000 | 0.000 | 8 | 8 | 1.000 | 0.000 | 0.000 |
| sssd08-04persp | 0.000 | 0.000 | 20080 | 17359 | 1.000 | 0.000 | 0.000 |
| sssd12-05persp | 0.131 | 0.133 | 63030 | 73358 | 0.016 | 0.016 | 0.154 |
| sssd15-04persp | 0.188 | 0.181 | 76121 | 77773 | -0.041 | -0.039 | -0.018 |
| sssd15-06persp | 0.285 | 0.260 | 43387 | 47623 | -0.095 | -0.087 | 0.002 |
| sssd15-08persp | 0.235 | 0.234 | 30374 | 41340 | -0.005 | -0.005 | 0.261 |
| sssd16-07persp | 0.232 | 0.214 | 41858 | 46101 | -0.086 | -0.079 | 0.014 |
| sssd18-06persp | 0.200 | 0.188 | 40346 | 48551 | -0.063 | -0.059 | 0.117 |
| sssd18-08persp | 0.383 | 0.372 | 31676 | 41841 | -0.028 | -0.027 | 0.221 |
| sssd20-04persp | 0.202 | 0.202 | 63012 | 69887 | 0.002 | 0.002 | 0.100 |
| sssd20-08persp | 0.202 | 0.195 | 27951 | 34670 | -0.036 | -0.035 | 0.164 |
| sssd22-08persp | 0.228 | 0.212 | 31593 | 33795 | -0.077 | -0.071 | -0.007 |
| sssd25-08persp | 0.178 | 0.172 | 27400 | 33837 | -0.038 | -0.037 | 0.159 |
| st_bsj2 | 0.000 | 0.000 | 17 | 15 | 1.000 | 0.000 | 0.000 |
| st_e05 | 0.000 | 0.000 | 59 | 75 | 1.000 | 0.000 | 0.000 |
| st_e24 | 0.000 | 0.000 | 7 | 7 | 1.000 | 0.000 | 0.000 |
| st_e25 | 0.000 | 0.000 | 15 | 15 | 1.000 | 0.000 | 0.000 |
| st_e30 | 0.000 | 0.000 | 47 | 61 | 1.000 | 0.000 | 0.000 |
| st_e31 | 0.000 | 0.000 | 593 | 490 | 1.000 | 0.000 | 0.000 |
| st_fp7a | 0.000 | 0.000 | 297 | 345 | 1.000 | 0.000 | 0.000 |
| st_fp7b | 0.000 | 0.000 | 349 | 341 | 1.000 | 0.000 | 0.000 |
| st_fp7c | 0.000 | 0.000 | 253 | 449 | 1.000 | 0.000 | 0.000 |
| st_fp7d | 0.000 | 0.000 | 277 | 355 | 1.000 | 0.000 | 0.000 |
| st_fp7e | 0.000 | 0.000 | 1605 | 1831 | 1.000 | 0.000 | 0.000 |
| st_fp8 | 0.000 | 0.000 | 69 | 63 | 1.000 | 0.000 | 0.000 |
| st_glmp_ss1 | 0.000 | 0.000 | 23 | 25 | 1.000 | 0.000 | 0.000 |
| st_ht | 0.000 | 0.000 | 13 | 11 | 1.000 | 0.000 | 0.000 |
| st_iqpbk1 | 0.000 | 0.000 | 37 | 37 | 1.000 | 0.000 | 0.000 |
| st_iqpbk2 | 0.000 | 0.000 | 39 | 37 | 1.000 | 0.000 | 0.000 |
| st_jcbpaf2 | 0.000 | 0.000 | 9 | 13 | 1.000 | 0.000 | 0.000 |
| st_m1 | 0.000 | 0.000 | 783 | 383 | 1.000 | 0.000 | 0.000 |
| st_m2 | 0.000 | 0.000 | 637 | 619 | 1.000 | 0.000 | 0.000 |
| st_pan1 | 0.000 | 0.000 | 11 | 11 | 1.000 | 0.000 | 0.000 |
| st_ph11 | 0.000 | 0.000 | 11 | 11 | 1.000 | 0.000 | 0.000 |
| st_ph12 | 0.000 | 0.000 | 13 | 13 | 1.000 | 0.000 | 0.000 |
| st_ph13 | 0.000 | 0.000 | 9 | 9 | 1.000 | 0.000 | 0.000 |
| st_qpc-m1 | 0.000 | 0.000 | 15 | 17 | 1.000 | 0.000 | 0.000 |
| st_qpc-m3a | 0.000 | 0.000 | 1269 | 1291 | 1.000 | 0.000 | 0.000 |
| st_qpk1 | 0.000 | 0.000 | 7 | 7 | 1.000 | 0.000 | 0.000 |
| st_qpk2 | 0.000 | 0.000 | 27 | 27 | 1.000 | 0.000 | 0.000 |
| st_qpk3 | 0.000 | 0.000 | 137 | 133 | 1.000 | 0.000 | 0.000 |
| st_rv1 | 0.000 | 0.000 | 107 | 81 | 1.000 | 0.000 | 0.000 |
| st_rv2 | 0.000 | 0.000 | 133 | 119 | 1.000 | 0.000 | 0.000 |
| st_rv3 | 0.000 | 0.000 | 511 | 629 | 1.000 | 0.000 | 0.000 |
| st_rv7 | 0.000 | 0.000 | 1143 | 1153 | 1.000 | 0.000 | 0.000 |
| st_rv8 | 0.000 | 0.000 | 1047 | 1269 | 1.000 | 0.000 | 0.000 |
| st_rv9 | 0.000 | 0.000 | 3349 | 1875 | 1.000 | 0.000 | 0.000 |
| st_testgr1 | 0.000 | 0.000 | 38 | 21 | 1.000 | 0.000 | 0.000 |
| st_z | 0.000 | 0.000 | 9 | 9 | 1.000 | 0.000 | 0.000 |
| supplychain | 0.000 | 0.000 | 119 | 95 | 1.000 | 0.000 | 0.000 |
| tln12 | 0.295 | 0.217 | 20517 | 22942 | -0.362 | -0.266 | -0.179 |
| tln4 | 0.000 | 0.000 | 13 | 25 | 1.000 | 0.000 | 0.000 |
| tln6 | 0.000 | 0.000 | 40 | 38 | 1.000 | 0.000 | 0.000 |
| tln7 | 0.075 | 0.121 | 52425 | 60523 | 0.375 | 0.375 | 0.457 |
| toroidal3g7_6666 | 0.200 | 0.117 | 51 | 213 | -0.706 | -0.414 | 0.592 |
| tricp | $\infty$ | $\infty$ | 275 | 356 | 0.000 | 0.000 | 0.228 |
| util | 0.000 | 0.000 | 48 | 38 | 1.000 | 0.000 | 0.000 |
| wastewater02m1 | 0.000 | 0.000 | 43 | 43 | 1.000 | 0.000 | 0.000 |
| wastewater02m2 | 0.000 | 0.000 | 35 | 31 | 1.000 | 0.000 | 0.000 |
| wastewater04m1 | 0.000 | 0.000 | 117 | 81 | 1.000 | 0.000 | 0.000 |
| wastewater04m2 | 0.000 | 0.000 | 25 | 25 | 1.000 | 0.000 | 0.000 |
| wastewater05m1 | 0.000 | 0.000 | 2561 | 3047 | 1.000 | 0.000 | 0.000 |
| wastewater05m2 | 0.000 | 0.000 | 4068 | 7429 | 1.000 | 0.000 | 0.000 |
| wastewater11m1 | 0.116 | 0.131 | 40219 | 43385 | 0.113 | 0.113 | 0.177 |
| wastewater11m2 | 0.385 | 0.431 | 15161 | 15304 | 0.106 | 0.106 | 0.114 |
| wastewater12m1 | 0.099 | 0.045 | 23070 | 28082 | -1.000 | -0.541 | -0.440 |
| wastewater12m2 | 0.460 | 0.654 | 7232 | 7822 | 0.296 | 0.296 | 0.349 |
| wastewater13m1 | 0.446 | 0.370 | 12150 | 16381 | -0.207 | -0.171 | 0.105 |
| wastewater13m2 | 0.538 | 0.538 | 6204 | 6129 | 0.000 | 0.000 | -0.012 |
| wastewater14m1 | 0.151 | 0.122 | 38064 | 42510 | -0.236 | -0.191 | -0.096 |
| wastewater14m2 | 0.191 | 0.209 | 11743 | 13355 | 0.084 | 0.084 | 0.194 |
| wastewater15m1 | 0.000 | 0.000 | 7735 | 8130 | 1.000 | 0.000 | 0.000 |
| wastewater15m2 | 0.000 | 0.000 | 54228 | 59163 | 0.982 | -0.000 | -0.000 |
| watercontamination0303 | 0.000 | 0.000 | 9 | 9 | 1.000 | 0.000 | 0.000 |
| watercontamination0303r | $\infty$ | $\infty$ | 22 | 37 | 0.000 | 0.000 | 0.405 |
| waterund01 | 0.000 | 0.000 | 49001 | 57176 | -0.022 | -0.021 | 0.056 |
| waterund08 | 0.000 | 0.000 | 38355 | 41489 | 0.335 | 0.083 | 0.003 |

| Name | Gap Ours | Gap Base | Nodes Ours | Nodes Base | Reward | Utility | Utility/Node |
|------|----------|----------|------------|------------|--------|---------|--------------|
| waterund11 | 0.001 | 0.001 | 35021 | 40695 | -0.736 | -0.420 | -0.241 |
| waterund14 | 0.009 | 0.009 | 9789 | 10684 | -0.012 | -0.012 | 0.072 |
| waterund17 | 0.001 | 0.001 | 35708 | 36527 | 0.549 | 0.545 | 0.436 |
| waterund18 | 0.001 | 0.001 | 34080 | 36286 | 0.049 | 0.048 | 0.085 |
| waterund22 | 0.016 | 0.017 | 10195 | 10702 | 0.016 | 0.016 | 0.062 |
| waterund25 | 0.080 | 0.094 | 11100 | 10382 | 0.154 | 0.153 | 0.095 |
| waterund27 | 0.089 | 0.089 | 2253 | 2835 | 0.001 | 0.001 | 0.206 |
| waterund28 | 0.080 | 0.080 | 18 | 17 | 0.000 | 0.000 | -0.056 |
| waterund36 | 0.100 | 0.082 | 1841 | 2443 | -0.217 | -0.178 | 0.083 |
| Mean | — | — | 6315 | 7463 | **0.487** | 0.000 | **0.114** |

## J  Alternative value function estimation

| Benchmark | Reward | Win-rate | geo-mean Ours | geo-mean SCIP |
|-----------|--------|----------|---------------|---------------|
| TSPLIB (Reinelt, 1991) | 0.150 | 0.68 | 0.86 | 0.957 |
| UFLP (Kochetov & Ivanenko, 2005) | -0.071 | 0.39 | 0.569 | 0.552 |
| MINLPLib (Bussieck et al., 2003) | 0.497 | 0.84 | 32.5 | 31.185 |
| MIPLIB (Gleixner et al., 2021) | 0.038 | 0.61 | 789.31 | 848.628 |
| TSPLIB@5min | 0.173 | 0.63 | 1.73 | 2.000 |
| MINLPlib@5min | 0.507 | 0.82 | 18.04 | 20.460 |
| MIPLIB@5min | 0.282 | 0.77 | 44.892 | 106.400 |

We compare a different value function $V(s)$ parametrization, which can be interpreted as the value function as on-policy q-values $V(s) = \int \pi(a|s)Q(s,a)da$ and the fact that our actions correspond to individual nodes:

$$Q(n|s) = \frac{\tilde{Q}(n|s)}{|P(r,n)|} \tag{18}$$

$$\tilde{Q}(n|s) = \tilde{Q}(\text{left}|s) + \tilde{Q}(\text{right}|s) + q(h_K(n)|s) \tag{19}$$

$$V(s) = \sum_{\forall n \in \mathscr{C}} \tilde{Q}(n)\pi(n|s)dn \tag{20}$$

where $q(h_n)$ are the learned per-node estimator, $\tilde{Q}$ the unnormalized Q-value, and $\mathscr{C}$ is the set of open nodes as proposed by the branch-and-bound method. The advantage of this interpretation is that it allows one to interpret the path estimates $\tilde{Q}$ as the on-policy q-values for each path. Having access to the q-values may be necessary for different actor-critic methods or the use of Deep Q-learning (Mnih et al., 2013).

## K  Pseudocode

---
**Algorithm 1** Pseudocode for node selection using our policy
---
1: **procedure** NODE SELECTOR
2:    Apply message passing over tree $T$ using Eq. 7
3:    Compute root-to-leaf paths using Eq. 8
4:    Compute policy $\pi$ over selectable leaves using Eq. 9 ▷ The value-function is only needed for training
5:    Sample selected node $n \sim \pi$
6:    run SCIP on $n$ to get new children $c_1, c_2$
7:    append $c_1$ and $c_2$ as children to $n$ in tree $T$
8:    (if necessary) prune $T$ according to SCIP pruning rules
---

For training, we set $N_{iter} = 16$, $K = 128$, and $\varepsilon = 0.1$, and the weight of the entropy regularization $\lambda_{ent} = 0.01$. In general, the policy improvement is the original PPO loss proposed in (Schulman et al., 2017), combined with implementation improvements provided in (Huang et al., 2021; 2022).

---
**Algorithm 2** Pseudocode for training our policy

---
1: **procedure** POLICY_EVALUATION($\pi$, )
2:      sample random instance according to Section 4.4
3:      Solve using SCIP for 45s and node selector Alg. 2, record all trees $\{T_i\}_0^N$, selected nodes $\{a_i\}_0^N$, and final optimality gap
4:      Solve using SCIP for 45s without our node selector and record the final gap
5:      compute the reward $r$ using Eq. 6
6:      compute the returns $R(s_t) = \gamma^{N-t} r$
7:      Compute Advantage $\{\hat{A}_i\}_0^N$ using GAE (Schulman et al., 2015) for all $\{T_i\}_0^N$ using value from Eq. 12.
     **return** $\{T_i\}_0^N$, $\{a_i\}_0^N$, $\{\hat{A}_i\}_0^N$, $R(s_t)$

8: **procedure** POLICY_IMPROVEMENT (SCHULMAN ET AL., 2017)($\pi$,$\{T_i\}_0^N$, $\{a_i\}_0^N$, $\{\hat{A}_i\}_0^N$, $R(s_t)$)
9:      copy policy function $\pi$ to $\pi_{\text{old}}$
10:      copy value function $V$ to $V_{\text{old}}$
11:      **for** iteration=$i \ldots N_{iter}$ **do**
12:          Optimize PPO-clip loss

$$L_t(\pi) = \min\left(\frac{\pi(a_i|T_i)}{\pi_{\text{old}}(a_i|T_i)}\hat{A}_i, \text{clip}\left(\frac{\pi(a_i|T_i)}{\pi_{\text{old}}(a_i|T_i)}, 1-\varepsilon, 1+\varepsilon\right)\hat{A}_i\right) + \lambda_{ent}H(\pi)$$

13:          using minibatch of size $K$ sampled from the exploration data.
14:          Optimize Value estimate using clippe value-estimation

$$mean = (R(s_t) - V(s_t))^2$$
$$clipped = V_{\text{old}}(s_t) + \text{clip}(V_{\text{old}}(s_t) - V(s_t), -\varepsilon, +\varepsilon)$$
$$L_v = \max(mean, clipped)$$

15:          using minibatch of size $K$ sampled from the exploration data.
     **return** $\pi$, $V$

16: **procedure** TRAIN($\pi$, V)
17:      **for** Iterations **do**
18:          $\{T_i\}_0^N$, $\{a_i\}_0^N$, $\{\hat{A}_i\}_0^N$, $R(s_t) \leftarrow$ Policy_Evaluation($\pi$)
19:          $\pi$, $V \leftarrow$ Policy_Improvement($\{T_i\}_0^N$, $\{a_i\}_0^N$, $\{\hat{A}_i\}_0^N$, $R(s_t)$)
     **return** $\pi$, $V$

---

