# OpenReview forum: "Reinforcement Learning for Node Selection in Branch-and-Bound"
_TMLR — Accepted by TMLR_

### Review · Reviewer_W5VF · 2024-08-30

**Summary Of Contributions:**

The authors present a work using reinforcement learning to learn policies for node selection in branch-and-bound algorithms for Mixed Integer Programming problems. Their technique improves on existing baselines, including the common software package SCIP as well as the baseline of Labassi et al. Rather than requiring a new optimisation for each problem instance, the technique allows pre-training and out-of-distribution performance that goes beyond other techniques that are problem-specific. It also allows for incorporating time consumption as an objective due to allowing rewarding low computational cost.

**Audience:**

Yes

**Broader Impact Concerns:**

No ethical conerns raised.

**Claims And Evidence:**

Yes

**Requested Changes:**

Section 1 to 2

“For simplicity, we focus our explanation to the case of mixed integer linear programs (MILP) while our method theoretically works for any type of constraint allowed in SCIP”

--> This is a comment about theory but refers to a software package. It would be more suitable to describe what kind of constraints are allowed.

“If a node has an upper bound larger than a currently found integral solution, no
node in that subtree has to be processed.”

I believe this should be the lower bound rather than the upper bound. If a node has a lower bound larger than a previously found solution, there is no point in checking that node (in a minimisation problem) since it cannot be the optimum.


The optimal diving oracle is mentioned many times but is never introduced or even paraphrased.

Section 3

 “Another frequently learned heuristic are cut-selection methods” --> Other frequently learned heuristics are cut-selection methods

“Cut selection is part of an extension of branch-and-bound
known as branch-and-cut, which additionally adds constraints to sub-problems.”

I think you should mention what kind of constraints and what their purpose is.


Section 4

The graph, and the corresponding actions, are dynamically changing. This approach may be computationally prohibitive for large graphs.

Section 4.1

The use of text in eq. 6 and 14  in Section 4.1 is not particularly, aesthetic, and scip should be capitalised.

Section 4.2,
“For processing the tree T , we use message-passing from the children to the parent.”

--> from parent to its children


Section 4.3:
The description of PPO does not seem correct and is incomplete. Since PPO formulates stochastic policies, the value function is typically the expected Q-value over the policy (not the maximum Q-value over states). Also there is no mention of the typically used generalized advantage estimation (GAE) scheme.  Looking at the code, the TreeList.py file includes the line
vds.append(max(vdict.values()))

so I believe the mistake is also in the experiments and the evaluation is actually not on-policy.
Consequently, the authors may get a better performance if they redo the experiments with V_pi rather than V*. I see that the code uses GAE (which is standard in PPO) so the authors can also mention this in the text.

Section 5

In Table 1, the authors choose to evaluate based on a particular time limit. Is there no way to evaluate this in terms of a more hardware and software independent manner? The number of nodes for instance. However, I understand that one of the claimed benefits of the RL technique is also allowing time optimisation.

the caption of Table 1 can be improved. The last two columns represent comparisons of the geometric mean of your method vs SCIP. However, the mean of what? And what do the other columns represent? A little bit more info can be found in the text but tables should be self-contained.

Section 5.4: the authors find that their method allows improved performance with a single network trained out-of-distribution, while Labassi et al.  use different networks train on the specific problem instance.

labassi --> Labassi


Section 5.5 presents results for interpreting the policies, and use an interpretability algorithm to this end.

General comment: the results do not contain any statistics for the spread (e.g. standard deviation).


References
Bestuzheva 2021a and Bestuzheva 2021b are identical references.


Appendix B:
Eq. 13 seems to be somewhat out of place. It is written just below a text where the authors talk about evaluating all paths, but actually refers to a single path. It may be more suitable to give the equation for log probabilities below the equation for p(leaf).


Appendix G:
Mathmatically --> Mathematically

**Strengths And Weaknesses:**

Strengths:

The technique is novel and well-designed as far as I can assess.

The authors obtain a consistently high performance on a variety of problem instances without the need to retrain.


Weaknesses:

The writing can be improved in some cases, primarily in terms of better introducing terms (see requested changes for concrete suggestions).

There is an error in the definition of the advantage function in the experiments. That is, the value used for its computation is using the maximal Q-value rather than the Q-value expected under the policy. This is not on-policy and is not in so it seems like experiments may need to be rerun with the correct definition. On the positive side, that means that the results obtained might be even better after rerunning them.

---

> ### Author Response · Authors · 2024-10-08
>
> We first want to thank the reviewer for his swift and very thorough analysis of our work. We especially want to thank for the acknowledgement of the generalization strength and novelty of our work, but also want to thank for the thorough reading of our appendix, which goes above and beyond the work expected by a reviewer.
>
> >There is an error in the definition of the advantage function in the experiments. That is, the value used for its computation is using the maximal Q-value rather than the Q-value expected under the policy. This is not on-policy and is not in so it seems like experiments may need to be rerun with the correct definition. On the positive side, that means that the results obtained might be even better after rerunning them.
>
> Thank you for pointing out that error, we re-run our main experiments on TSPlib, MINLPlib, MIPlib and UFLP, with the new results being found in Appendix J. Overall the results are slightly different, but reach about the same quality. This may seem counterintuitive at first, but makes sense if one remembers that PPO does not use TD-learning to compute its value function:
> PPO updates its value function on-policy using
>
> $$\min E[(V(s) – R(s))^2]$$
>
> Where R(s) are the monte-carlo returns sampled from the environment. Since this parametrization does not use any recursive estimation, one can view our “max over q-function” definition as simply an atypical parametrization of $V(s)$, i.e. our value function is parameterized as the maximum over $V(s) = \max(f_1(x), f_2(x),\dots,f_n(x))$. Of course, the interpretation of $f_i(x)$ as the on-policy q-value associated with action $i$ is no longer sound, but a different parametrization of the estimator $V(s)$ doesn’t change whether on- or off-policy estimation is done. The max is simply a permutation-invariant aggregator function.
> We rewrote our main text definition of the value function as a more general aggregation over $f_i$ functions, and note in the appendix that one can rewrite the value function to get on policy q-values by using a weighted integral rather than a max-aggregation. For our work, this doesn’t make a difference since we never use the q-functions themselves, but this may be useful in case someone wants to use e.g. Deep Q-learning for training this type of model that needs to parameterize the Q-functions explicitly.
>
>
> >“For simplicity, we focus our explanation to the case of mixed integer linear programs (MILP) while our method theoretically works for any type of constraint allowed in SCIP”
> --> This is a comment about theory but refers to a software package. It would be more suitable to describe what kind of constraints are allowed.
>
> This is true. The reason we reference the SCIP MINLP implementation is because precisely defining what kinds of constraints are allowed itself is a hard question to answer: In theory, any optimization problem that can be decomposed into a finite set of solvable sub-optimization problems can be solved using our method. However, defining this more accurately ends up being highly technical and up to constant changes: For instance, SCIP can solve any problem for which a “nonlinear handler” exists, which then, in turn, implements domain-propagation, separation, and branching. This means that not only are there huge and hard to describe sets of constraints supported by SCIP, but also that one could seamlessly add new constraint handlers to SCIP without affecting our method.
> Therefore we opted to simply refer to the SCIP implementation since that gives the most accurate view on the types of constraints that could be handled by our method.
>
>
>
> >“If a node has an upper bound larger than a currently found integral solution, no node in that subtree has to be processed.”
> I believe this should be the lower bound rather than the upper bound. If a node has a lower bound larger than a previously found solution, there is no point in checking that node (in a minimisation problem) since it cannot be the optimum.
> The optimal diving oracle is mentioned many times but is never introduced or even paraphrased.
>
> We have updated the paper to include a definition of diving oracles, and fixed the lower-bound upper-bound error.

---

> > ### Author Response · Authors · 2024-10-08
> >
> > >“Cut selection is part of an extension of branch-and-bound known as branch-and-cut, which additionally adds constraints to sub-problems.” I think you should mention what kind of constraints and what their purpose is.
> >
> > We have added a clearer explanation for cuts and cut-selection.
> >
> > > The graph, and the corresponding actions, are dynamically changing. This approach may be computationally prohibitive for large graphs.
> >
> > This is true, but considering that our model is rather small one could scale quite dramatically if the method wasn’t implemented in python. More generally, there is a tension between making the model large, which increases its representational capabilities and (empirically) makes training easier, and making the models fast. One additional aspect we did not explore is moving the node-selection estimator to the GPU: We didn’t do this to make the comparisons against existing node-selectors fair. Further, modern techniques such as pruning or quantization could be used to further speed up inference.
> >  The bigger limitation is the need to train our method on larger graphs, which usually involves training on larger instances. This makes training prohibitively expensive for us, which is why our training times are limited at 45s.
> >
> > >Section 4.2, “For processing the tree T , we use message-passing from the children to the parent.” --> from parent to its children
> >
> > The message-passing is from children to parent, but the path-aggregation is from root to leaf: The motivation behind this is that to assess the quality of a node $n$ we look at the subtree starting from that node. This aligns with prior work on node-selection (e.g., “A Study of Learning Search Approximation in Mixed Integer Branch and Bound: Node Selection in SCIP”, Yilmaz et al. 2022), where the quality of the left and right children is explicitly encoded into the features. Since our work already uses a branch-and-bound graph, we can generalize this "child-aggregation" using a learned message-passing layer. $K$-message passing iterations corresponds to the K-depth descendents of the original node.
> > After the subtree-features are aggregated, we compute the root-to-leaf paths to get the weight of each leaf node.
> >
> > >In Table 1, the authors choose to evaluate based on a particular time limit. Is there no way to evaluate this in terms of a more hardware and software independent manner? The number of nodes for instance. However, I understand that one of the claimed benefits of the RL technique is also allowing time optimisation.
> >
> > There is no good way of estimating node-selection performance in a hard- and software independent manner. Node selection (especially in the nonlinear case) depends on the abilities of the solver itself, since what constitutes a “good” node is highly dependent on the specifics of cut-selectors, primal heuristics, diving heuristics, etc.
> > Simply counting nodes is insufficient to estimate performance, since the processing time each node takes can be substantially different. It is not uncommon to have two nodes, one which reduces the duality gap by a lot, the other by only a little, but the first node is substantially more expensive than the second. Therefore, node selection has the implicit problem of both estimating the quality of the node, but also estimating the cost of evaluating the node.
> > In MINLPlib we find that the effect of different nodes having different costs is particularly pronounced.
> >
> > >General comment: the results do not contain any statistics for the spread (e.g. standard deviation).
> >
> > We have added standard deviations to the Rewards in Table 1. Since the geometric means are far from being normally distributed and have _very_ long tails due to some problems being substantially harder than others. “Reward” which normalizes based on problem complexity, does not have this problem.
> >
> >
> > We have updated our work and highlighted the changed parts in red.

---

### Review · Reviewer_4fth · 2024-09-24

**Summary Of Contributions:**

The paper proposes a novel application of reinforcement learning to the problem of node selection in branch-and-bound algorithms. The proposed method builds a graph neural network that is similar in graph structure to the branch-and-bound tree, capturing the properties of individual nodes as well as the tree structure. The model is trained on a carefully constructed set of intermediate-difficulty traveling salesman problems and then tested on multiple different benchmarks, including benchmarks comprised of different instances of the same problem, and heterogeneous benchmarks containing mixed-integer linear and nonlinear problems. Within the time limits used in the paper, the proposed method demonstrates considerable improvements and good capabilities for generalization to adjacent problem classes.

**Audience:**

Yes

**Broader Impact Concerns:**

I have no broader impact concerns.

**Claims And Evidence:**

Yes

**Requested Changes:**

- While the paper states its limitations (that stem from limited computational resources) quite clearly, I would suggest that they are also mentioned in the introduction and/or abstract, for example the fact that the training and tests are conducted with small time limits.
- Figure 1: acronym MLP undefined.
- Page 3, second to last paragraph: another paper proposing an ML-based cut selection method is "Adaptive Cut Selection in Mixed-Integer Linear Programming" by Turner et al.
- Equation 5: a full definition of the notation used in the equation would help readability, especially for readers not used to this notation.
- Equation 6: does gap(scip) denote the gap achieved by default SCIP after making one node selection decision from the same state of the tree? A clarification of the definition would be helpful here.
- In Section 5.2.1 and below, there are a few misspellings of the word 'lose'.
- Page 10, third to last paragraph: 'seperate' should be 'separate'.
- What was the exact setup of the experiments in Section 5.4?

**Strengths And Weaknesses:**

Strengths:

- The method is well-designed, improving upon previously existing methods in terms of its ability to utilize information about the branch-and-bound tree, and specifics of the algorithm such as reward functions or the setup for applying reinforcement learning are well thought out, addressing some of the challenges remaining in previous works.
- The experiments are thorough in terms of the variety of the benchmarks used, convincingly proving that the algorithm generalizes well beyond the TSP instances it was trained on.
- The paper includes an interesting discussion of the insights drawn from the trained model, which, on the one hand, are in line with what is already known by the optimization community, and, on the other hand, provide new information on what node and tree properties may be relevant.

Weaknesses:
- The short time limits impose a limitation on the conclusions one can make regarding the method's performance for solving challenging instances to optimality or small gaps, which is one of the main tasks in optimization research, as improvements on such instances mean pushing the boundaries of what problems algorithms can solve. The paper does mention this weakness, and it attempts to somewhat alleviate it by arguing that the results generalize well from 45s to 5min. However, with longer runtimes solvers may shift into different 'modes' of solving, for instance, when a good feasible solution is found and what remains is proving optimality, and what works well for the initial few minutes might not perform well when the solver is run for an hour or two. I understand, however, that learning, and learning in the context of branch-and-bound, is a very computationally expensive task, and these limitations are out of the authors' control.

---

> ### Author Response · Authors · 2024-10-08
>
> We thank the reviewer for his through analysis of our paper and particularly highlighting the strengths of our analysis and benchmarking.
> First and foremost, regarding the short timelimit: This is indeed a limitation of ours method (and to our knowledge all other learned node selectors). We tried our best to estimate the impact of the reduced training time, but we are fundamentally limited from a computational point of view.
>
> >  While the paper states its limitations (that stem from limited computational resources) quite clearly, I would suggest that they are also mentioned in the introduction and/or abstract, for example the fact that the training and tests are conducted with small time limits.
>
> We have added that training is done for short time limits of 45s into the abstract
>
>
> >Figure 1: acronym MLP undefined.
>
> We have added the definition of “MLP” to the figure description.
>
> >  Page 3, second to last paragraph: another paper proposing an ML-based cut selection method is "Adaptive Cut Selection in Mixed-Integer Linear Programming" by Turner et al.
>
> We have added a citation to Turner et al.’s work.
>
> >Equation 5: a full definition of the notation used in the equation would help readability, especially for readers not used to this notation.
>
> We have added some additional comments regarding notation.
>
> >  Equation 6: does gap(scip) denote the gap achieved by default SCIP after making one node selection decision from the same state of the tree? A clarification of the definition would be helpful here.
>
> We have added a better description of the gap(…) notation: The gap of both methods is evaluated at the end of the time limit, meaning our method only receives a reward at the last timestep.
>
> >  In Section 5.2.1 and below, there are a few misspellings of the word 'lose'.
>
> Fixed the misspellings
>
> >  Page 10, third to last paragraph: 'seperate' should be 'separate'.
>
> Thank you, is corrected.
>
> >  What was the exact setup of the experiments in Section 5.4?
>
> We used the instances by labassi et al and ran our method against labassi’s according to the evaluation scheme proposed in their work, which amounts to measuring the time and number of nodes needed to solve the instances to completion. For this we use the best performing models provided by labassi on their “transfer”-test set.
>
>
> Thank you very much for your analysis of our work. We have fixed all your points in our new version and highlighted the updated parts in red.

---

> ### Comment · Reviewer_4fth · 2024-10-18
> **Response to revision**
>
> Most of my comments have been fully addressed in the revision. In the paragraph above equation (5), I suggest including an introduction of the notation $x|y$ used in the function arguments in (5).
>
> I have noticed a few new typos and misspellings in the newly added/changed text. For instance:
> - The sentence starting with 'Other frequently learned heuristic' (last paragraph on page 3) incorrectly uses the singular form of 'heuristic', and could generally be rephrased.
> - 'solutions that are feasible in a nodes...' (last paragraph on page 3) should be 'solutions that are feasible in a node's...'
>
> There are more misspellings, therefore I recommend that the authors carefully proofread the new text.
>
> Finally, and unrelated to the content itself, I would advise that the authors use gender-neutral pronouns to refer to reviewers.

---

> > ### Author Response · Authors · 2024-10-19
> >
> > Thank you for your feedback and further suggestions: We have fixed the misspellings and updated the paragraph above equation (5) to more clearly define the notation of the conditional used in $\pi(a|s)$.
> >
> > These fixes are already included in the new version we have uploaded just now.

---

### Review · Reviewer_EJge · 2024-10-17

**Summary Of Contributions:**

This work uses (Reinforcement Learning) RL to train a Graph Neural Network (GNN) to learn the policy for branch-and-bound (BnB) node selection for solving mixed-integer (nonlinear) programming problems. The GNN captures the features inside the BnB tree structure and produces the probability of selecting nodes. The experiments demonstrate the superior performance of the proposed method compared to classical solver SCIP and a state-of-the-art learning method on multiple benchmark datasets.

**Audience:**

Yes

**Claims And Evidence:**

Yes

**Requested Changes:**

1. There are abbreviations (e.g., GNN) but no full names, please check the paper to correct it.
2. A summary of contributions in the introduction may help the reader get the points better.
3. It is recommended to include a pseudocode for the whole framework to help the reader understand the proposed method easier.
4. The tenses of the articles need to be consistent. For example, in Section 5.5 Ablations, the first and second paragraphs use inconsistent tenses.
5. "4.1 Reward definition" -> "4.1 Reward Definition".

**Strengths And Weaknesses:**

Strengths:
1. Clear motivation and good presentation.
2. The idea of the proposed method is intuitive and simple, and the experimental results fully demonstrate the proposed method's promising performance.

Weaknesses:
1. More learning-based methods can be included for comparison.
2. The feature selection for the model input can be optimized.

---

> ### Author Response · Authors · 2024-10-19
>
> First and foremost, we want to thank the reviewer for reading our paper on such short notice. We also want to thank the reviewer for noting the benchmarking and intuitiveness of our work.
> Regarding the weaknesses:
> >  More learning-based methods can be included for comparison.
>
> This is true, but unfortunately hard to do in practice. We were not able to reproduce the results by Yilmaz et al. and  He et al. This is because of both a lack of code/datasets, but also due to the age of these works: Yilmaz and He both use SCIP 6, compared to Ours which uses SCIP 8. Forward- or back-porting our work is quite challenging due to changes in the API between SCIP 6 and 7. SCIP has improved quite significantly in many ways, including node selection.
> Further, looking at Yilmaz et al.’s work (table 6, https://arxiv.org/pdf/2007.03948) one can see that neither Yilmaz nor He are able to beat SCIP 6’s basic node selector, even when trained and tested on exactly the same instance type. In their other experiments they are sometimes able to beat SCIP 6’s basic node selector, but those benchmarks are not really representative of real performance: Yilmaz et al. compare the different methods against a time limit which is set by picking the time where the ML policy achieved the lowest harmonic mean between the mean solving time and the mean optimality gap across the instances
>
> >The time limit for each instance is based on the solving time of the ML policy that achieved the lowest harmonic mean between the mean solving time and mean optimality gap across all instances
>
> (https://arxiv.org/pdf/2007.03948, p 163). This means that they compare their method’s optimal configuration (wrt time limit and optimality gap) against SCIP’s default selection. This comparison makes sense if one wants to benchmark the potential of learned node selectors but is of little practical relevance (unless one’s real time limit happens to be equal to the optimal node selection time…).
>
> >  The feature selection for the model input can be optimized.
>
> This is the definitely case. However, we argue that this is a somewhat orthogonal to our method as what features to include is a problem in all learned and even classical node selectors. The advantage of learned techniques is that they can “soft discard” unneeded information. To test this, we utilize explainable AI techniques (see appendix H) to detect the importance of individual features.
> However, we generally agree that better features would improve our method further (especially better features for nonlinear programming).
>
> >  There are abbreviations (e.g., GNN) but no full names, please check the paper to correct it.
>
> We will do that
>
> >  A summary of contributions in the introduction may help the reader get the points better.
>
> We can also do this
> >  It is recommended to include a pseudocode for the whole framework to help the reader understand the proposed method easier.
>
> We can do this, but I feel that this will not be insightful: the pseudocode would be close to
>
>       While not time elapsed:
>           node~pi(tree)
>           node1, node2 <- bnb_solve(node)
>           Tree.add_children(node, node1, node2)
>           Tree.prune()
> Since our method is mostly about finding a good parametrization of the ever-changing Branch-and-Bound tree. The actual node-selection algorithm is just about sampling one of the nodes from the learned distribution, combined with some book-keeping that updates the dynamic tree-representation.
> We have added pseudocode in the appendix
>
> >  The tenses of the articles need to be consistent. For example, in Section 5.5 Ablations, the first and second paragraphs use inconsistent tenses.
>
> We are going to fix this as well.
>
> > "4.1 Reward definition" -> "4.1 Reward Definition".
>
> We will fix the typo.
>
> Thank you once again for your short-notice replacement and insightful comment.

---

> > ### Comment · Reviewer_EJge · 2024-10-19
> >
> > Thanks to the author's reply, most of my concerns have been addressed. But I hope the author can provide pseudocode for the training and testing phases, not just the testing phase, so that readers can understand the entire algorithm framework more easily and quickly. Other than that, I have no other concerns.

---

> > > ### Author Response · Authors · 2024-10-22
> > >
> > > Dear Reviewer, we have added pseudocode for the training phase as well.
> > >
> > > The implementation of the PPO algorithm we use is very close to the original one proposed by Schulman, with a few modern "best practices" outlined in "The 37 Implementation Details of Proximal Policy Optimization" by Huang et al.
> > >
> > > Thank you for your time and quick response!

---

### Comment · Action_Editor_dq19 · 2024-10-14
**Update of the Current Review Process**

Dear Authors,

Thank you for submitting your work to TMLR, and sorry for the long waiting time after the review deadline. Here is the update of the current review process.

This paper was originally assigned to three reviewers, but only one review comment was obtained by the end of August. A new expert reviewer has been recruited, and a new review comment has been submitted by 25 Sept. I still have not received any response from two of the original reviewers by comment or email.

Therefore, one new emergent reviewer has been recruited, and the review comment will be available by this Friday. Once three review comments are obtained, the two unresponsive original reviewers will be removed from this work.

Best Regards,

AE

---

### Decision · Action_Editor_dq19 · 2024-11-28

**Recommendation:** Accept as is

**Comment:**

The reviewers find the proposed method is novel, well motivated and well-designed, and they also appreciate the thorough experiments that clearly show the promising performance of the proposed method. After rebuttal, most of the raised concerns were properly addressed, and one reviewer voted to accept this paper while two reviewers leaned toward accepting.

I read the paper in detail and totally agree with all reviewers that this work clearly meets the TMLR acceptance criteria (solid and well-supported claims, potential audience). In particular, it is promising to see a fixed policy pretrained on synthetic TSP instances can perform well on the heterogeneous instances in MINLPLib and MIPLIB. Therefore, I am happy to accept this work as is.

**Audience:**

All reviewers believe some individuals in TMLR's audience could be interested in the findings of this paper.

**Claims And Evidence:**

This work proposes a novel reinforcement learning (RL) based approach for branch-and-bound node selection. Unlike the existing learning-based approaches that only use individual node data, by training a graph neural network model (GNN), the proposed method leverages the entire tree state (e.g., the probability distribution from root to each node) to select the optimal node. Thorough experimental results show that, using the proposed method, a fixed policy only pretrained on synthetic traveling salesmen problem (TSP) instances can achieve promising performance on different instances in TSPLIB as well as the heterogeneous instances in MINLPLib and MIPLIB.

All reviewers believe the claims made in this paper are supported by accurate, convincing and clear evidence.